# JET-Diff: Joint-Encoding Tensor Diffusion Model for Accurate DTI Reconstruction from Sparse DWIs

## Abstract

Diffusion Tensor Imaging (DTI) is an advanced magnetic resonance imaging (MRI) technique for characterizing white matter microstructure. Conventional DTI protocols require multiple diffusion-weighted imaging (DWI) acquisitions across numerous directions, resulting in long scan times, motion artifacts, patient discomfort, and reduced clinical utility. Current deep learning approaches frequently yield diffusion tensors that are anatomically inconsistent or physically implausible. We introduce Joint-Encoding Tensor Diffusion (JET-Diff), a framework that synthesizes the full six-component diffusion tensor in 3D from a highly undersampled DWI acquisition. Specifically, we propose a latent diffusion model operating on a set of coupled latent tensors derived from sparse DWIs and diffusion tensor components, which improves anatomical fidelity and encourages physically consistent tensors. JET-Diff leverages a novel anatomical autoencoder to disentangle structural information from tensor properties, yielding a compact and expressive latent space optimized for generative performance. Experiments on the Human Connectome Project (HCP) Young Adult dataset demonstrate that JET-Diff improves reconstruction accuracy and produces geometrically consistent diffusion tensors, as evidenced by SPD-aware validity metrics such as Log-Euclidean and tractography-based distances.

## 1 Introduction

Diffusion Tensor Imaging (DTI) is a Magnetic Resonance Imaging (MRI) technique that quantifies anisotropic water diffusion, enabling non-invasive characterization of white matter microstructure (Basser et al., 1994; Le Bihan et al., 2001). It supports mapping of neural pathways and extraction of clinically relevant biomarkers across neurological disorders (Behrens et al., 2007; Andica et al., 2020). However, its clinical adoption is constrained by long acquisition times (Le Bihan et al., 2001). High-quality tensor estimation often requires more than thirty Diffusion-Weighted Images (DWIs) to adequately sample the diffusion signal (Mukherjee et al., 2008). This prolongs scans, which is a source of patient discomfort and a strain on clinical resources, and increases susceptibility to motion artifacts that degrade tensor accuracy (O'Donnell & Westin, 2011). Developing methods that can learn empirical priors to reconstruct reliable tensors from a substantially reduced number of DWIs could streamline routine scans, improving both efficiency and diagnostic accuracy.

Reconstructing the six independent components of the diffusion tensor from a sparse set of DWIs is a severely ill-posed inverse problem (Lenglet et al., 2009). Traditional fitting methods like least-squares are mathematically underdetermined and fail to produce reliable results. While deep generative models have emerged as a promising data-driven solution (Tian et al., 2020; Li et al., 2021; Zhang et al., 2024), existing approaches suffer from critical limitations that compromise anatomical fidelity and physical plausibility. Many models operate on a 2D, slice-by-slice basis, disregarding volumetric continuity of neural structures and leading to anatomical inconsistencies in reconstructed 3D volumes. Others directly synthesize DTI-derived parameter maps, such as fractional anisotropy (FA) or mean diffusivity (MD), but this bypasses reconstruction of the full diffusion tensor and does not explicitly address the physical constraints of diffusion imaging, since scalar maps are secondary quantities; moreover, anatomically plausible scalar maps may still correspond to geometrically inconsistent tensors. Most critically for latent-based models, autoencoders often produce entangled la-

tent representations, forcing a single information bottleneck to capture both fine-grained anatomical detail and complex tensor characteristics, which induces an inherent trade-off between compression efficiency and reconstruction fidelity (Higgins et al., 2017; Chen et al.).

To overcome these limitations, we propose Joint-Encoding Tensor Diffusion (JET-Diff), a framework for high-fidelity DTI synthesis from sparse measurements. In this work, we specifically focus on the practically relevant setting of an extremely sparse and fixed acquisition, aiming to learn an empirical prior that captures consistent relationships between sparse DWIs and their associated tensor fields under this acquisition configuration. The core idea is a latent diffusion model operating on a set of coupled latent tensors that represent the input DWI and output DTI components as a single unified entity in latent space. By learning such a coupled latent field, JET-Diff captures anatomical and microstructural relationships between sparse DWIs and their associated diffusion tensors.

Our framework is implemented as a carefully designed latent-diffusion pipeline. First, we introduce an Anatomical Autoencoder based on the principle of information decoupling. By providing anatomical context directly to the decoder, the latent space is freed to primarily encode essential tensor characteristics, yielding a more efficient and expressive representation. Second, a latent diffusion model is trained within this high-fidelity latent space to generate the complete tensor field from sparse DWI latents. By operating volumetrically within a disentangled latent space and applying diffusion over a coupled latent field, JET-Diff produces whole-brain diffusion tensor volumes that remain highly consistent with the input anatomy and exhibit improved symmetric positive definite (SPD)–aware tensor metrics, demonstrating measurable gains over existing diffusion tensor reconstruction methods.

## 2 RELATED WORK AND BACKGROUND

### 2.1 DIFFUSION TENSOR MODEL

Diffusion Tensor Imaging (DTI) is a foundational MRI technique that quantifies the anisotropic diffusion of water molecules in biological tissues, particularly the brain's white matter. The framework, introduced by Basser et al. (1994), models the diffusion process in each voxel using a $3 \times 3$ symmetric positive semi-definite tensor, $\mathbf{D}$. This tensor linearly relates the measured diffusion-weighted signal to the applied diffusion-sensitizing gradients, as described by the Stejskal-Tanner equation (Stejskal & Tanner, 1965):

$$S(\mathbf{g}) = S_0 \exp(-b\mathbf{g}^T\mathbf{D}\mathbf{g}),$$

where $S_0$ is the non-diffusion-weighted signal intensity, $b$ is the diffusion weighting factor, $\mathbf{g}$ is the diffusion gradient direction vector, and $\mathbf{D}$ is the diffusion tensor. This equation forms the physical basis for estimating the diffusion tensor from a series of diffusion-weighted measurements. Further details on the equation's parameters, tensor estimation, and derived metrics are provided in Appendix A.

Beyond tensor-based representations, diffusion MRI also includes higher-order models such as fiber orientation distributions (fODFs) (Tournier et al., 2007), spherical deconvolution, and sparse orientation modeling (Canales-Rodríguez et al., 2019). These methods target multi-shell or high-angular-resolution regimes and are primarily designed to resolve complex fiber configurations (Karimi & Warfield, 2024). In contrast, the present work focuses on full-tensor reconstruction under an extremely sparse single-shell acquisition, a setting for which high-order models are not typically applicable.

### 2.2 DTI RECONSTRUCTION FROM SPARSE ACQUISITIONS

The problem of reconstructing a diffusion tensor from an insufficient number of Diffusion-Weighted Images (DWIs) is a classic, ill-posed inverse problem (Tuch, 2004). Early approaches relied on linear or weighted linear least-squares fitting, which are computationally simple but highly unstable and sensitive to noise in low-signal regimes (Basser et al., 1994). Model-based approaches leveraged compressed sensing theory to exploit sparsity priors, with Knoll et al. (2015) introducing reconstruction that applied Total Variation constraints to preserve spatial coherence.

The advent of deep learning has substantially influenced DTI reconstruction. SuperDTI (Li et al., 2021) demonstrated that convolutional neural networks could directly map from sparse DWIs to

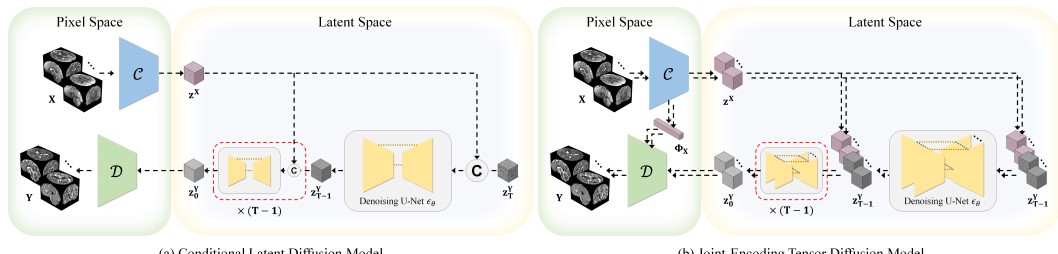

Figure 1: **Overview of the JET-Diff framework.** (a) A standard latent diffusion model applies the diffusion process to DTI latents $\{\mathbf{z_c^Y}\}$, conditioned on DWI latents $\{\mathbf{z_c^X}\}$ via concatenation. (b) JET-Diff applies the diffusion process to a coupled latent field that combines DWIs and DTIs, $\{\mathbf{z_c^X}, \mathbf{z_c^Y}\}$, enabling direct interactions between all components during denoising.

diffusion parameter maps, achieving strong reconstruction quality from as few as six gradient directions. FlexDTI (Wu et al., 2024) further incorporates gradient-direction flexibility, which is complementary to our fixed four-direction acquisition setting. However, most existing methods suffer from fundamental limitations: slice-wise processing ignores anatomical context, and direct synthesis of scalar maps independently can violate tensor-level consistency, as these metrics should derive from a single underlying tensor.

### 2.3 GENERATIVE DIFFUSION MODELS FOR MEDICAL IMAGING

Denoising Diffusion Probabilistic Models (DDPMs) (Sohl-Dickstein et al., 2015; Ho et al., 2020) have emerged as state-of-the-art generative models, with significant recent applications in medical imaging. These models have proven effective for tasks such as accelerated MRI reconstruction (Chung & Ye, 2022) and high-resolution 3D volume synthesis (Wang et al., 2025).

Within the domain of diffusion MRI, recent efforts have applied generative frameworks to denoising and reconstruction. Several self-supervised methods leverage diffusion models to restore signal quality from noisy acquisitions (Xiang et al.; Wu et al.). More directly related to our task, Diff-DTI (Zhang et al., 2024) was the first to employ a diffusion model for rapid DTI reconstruction from sparse DWIs. Its approach conditions the generative process on sparse DWI features to synthesize DTI-derived scalar maps like fractional anisotropy (FA) and mean diffusivity (MD). While Diff-DTI achieves impressive results, its reliance on an explicit guidance mechanism to generate secondary parameter maps bypasses the synthesis of the fundamental diffusion tensor. In contrast, our approach operates on coupled DWI and tensor latents and directly synthesizes the full six-component tensor, from which physically interpretable parameter maps are then calculated.

## 3 METHOD: JOINT-ENCODING TENSOR DIFFUSION (JET-DIFF)

This section details the Joint-Encoding Tensor Diffusion (JET-Diff) framework. We introduce a variant of the Latent Diffusion Model (LDM) (Rombach et al., 2022) instantiated as a latent diffusion model operating on a set of coupled latent tensors encoding sparse DWI inputs and their corresponding DTI fields, thereby promoting anatomical and tensor-level consistency.

### 3.1 PROBLEM DEFINITION AND OVERVIEW

The primary objective is to reconstruct a complete diffusion tensor field from a minimal set of Diffusion-Weighted Images (DWIs), which is a severely ill-posed inverse problem. More specifically, let the input $\mathbf{X}$ be the set of four DWI volumes, $\mathbf{X} = \{\mathbf{X_c}\}_{c=1}^4$, where each component $\mathbf{X_c} \in \mathbb{R}^{H \times W \times D}$ consists of one non-diffusion-weighted image ($b = 0$) and three DWI volumes. The desired output $\mathbf{Y}$ is the set of six diffusion tensor component volumes, $\mathbf{Y} = \{\mathbf{Y_c}\}_{c=1}^6$, where each component $\mathbf{Y_c} \in \mathbb{R}^{H \times W \times D}$ represents one of the unique tensor elements $(D_{xx}, D_{yy}, D_{zz}, D_{xy}, D_{xz}, D_{yz})$.

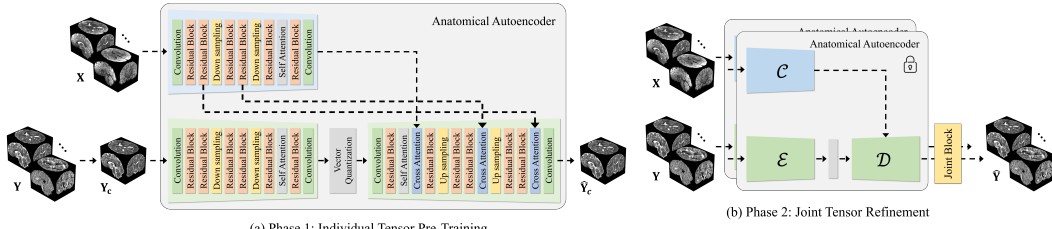

(a) Phase 1: Individual Tensor Pre-Training      (b) Phase 2: Joint Tensor Refinement

Figure 2: **Anatomical Autoencoder architecture and training.** (a) **Phase 1: Independent Pre-Training.** The encoder ($\mathcal{E}$) and conditioner ($\mathcal{C}$) respectively extract DTI latent codes and DWI anatomical features. The decoder ($\mathcal{D}$) reconstructs the DTI ($\hat{\mathbf{Y}}$) by fusing these via cross-attention. (b) **Phase 2: Joint Refinement.** With the main network frozen, a lightweight joint block is fine-tuned to promote cross-component consistency with negligible computational overhead.

Our proposed framework, JET-Diff, addresses this challenge with a latent-diffusion pipeline, illustrated in Figure 1. The first stage involves an Anatomical Autoencoder, composed of a tensor property encoder $\mathcal{E}$, an anatomical conditioner $\mathcal{C}$, and a DWI-aided decoder $\mathcal{D}$. This stage learns a compact latent representation of the tensor field, while ensuring anatomical consistency by explicitly conditioning the synthesis process on DWI features. The second stage employs a latent diffusion model that generates the tensor within this latent space by operating on a coupled latent field of DWI and tensor components.

Because the diffusion tensor is a deterministic fit to the full DWI set under the Stejskal-Tanner model (Stejskal & Tanner, 1965), the reference tensor $\hat{\mathbf{D}}$ in this work is treated simply as a deterministic function of the measured signal. JET-Diff does not model the physical diffusion process; instead, it learns an empirical latent prior that captures consistent relationships between the sparse DWI subset and the corresponding fitted tensors. Thus, "joint" refers to the coupled latent representation of DWI and tensor components rather than a probabilistic joint model.

## 3.2 Anatomical Autoencoder for High-Fidelity Latent Representation

The foundation of our generative framework is an autoencoder that maps high-dimensional tensor data into a compact latent space. The quality of this latent space is critical, as the performance of the subsequent diffusion model is bounded by the autoencoder's fidelity (Higgins et al., 2017; Chen et al.). Standard autoencoders are ill-suited for this task because they force a single bottleneck to encode both the tensor's physical properties and complex anatomical structure. This entangled representation is inefficient and prone to loss of fine details. Our Anatomical Autoencoder, depicted in Figure 2, addresses this limitation through a design centered on information decoupling. By implicitly separating anatomical context from tensor-specific information, the latent code is relieved from representing spatial structure and can focus on the intrinsic properties of the tensor field.

### 3.2.1 DWI-Aided Decoder for Information Decoupling

A key feature of our autoencoder is the principle of disentangling the latent representation of the tensor's properties (*what*) from its anatomical context (*where*). A conventional autoencoder must compress both into its latent code, creating a significant bottleneck that can lead to anatomical misalignment.

Our DWI-aided decoder, $\mathcal{D}$, resolves this by decoupling these responsibilities. The encoder $\mathcal{E}$ learns a highly efficient latent code $\mathbf{z}^{\mathbf{Y}}$ representing primarily the tensor's intrinsic properties. The anatomical context is extracted by a conditioner $\mathcal{C}$ directly from the input DWI stack $\mathbf{X}$ as a feature pyramid $\Phi_{\mathbf{X}} = \{\phi_{\mathbf{X}}^{l}\}_{l=1}^{L}$. During decoding, the decoder fuses the compact latent code $\mathbf{z}^{\mathbf{Y}}$ with these anatomical features $\Phi_{\mathbf{X}}$ at each resolution level. This fusion is achieved using cross-attention blocks that employ multi-axis cross attention (Tu et al., 2022) to maintain linear computational complexity, a critical requirement for processing high resolution medical images. This design allows the latent space to achieve a higher compression ratio while enabling the decoder to produce a final output $\hat{\mathbf{Y}} = \mathcal{D}(\mathbf{z}^{\mathbf{Y}}, \Phi_{\mathbf{X}})$ with high fidelity.

### 3.2.2 JOINT REFINEMENT FOR TENSOR CONSISTENCY

While the DWI-aided decoder ensures high fidelity for individual tensor components, it does not explicitly encourage the cross-component relationships required for a coherent tensor field. To address this, we introduce an efficient joint refinement phase within autoencoder pre-training, illustrated in Figure 2b. After the initial training, we freeze the weights of the conditioner $\mathcal{C}$, the encoder $\mathcal{E}$, and the majority of the decoder $\mathcal{D}$. We then insert a lightweight joint MLP block into the final layers of the decoder to promote interactions across the six tensor components. By fine-tuning only this joint block and the final convolution, thereby encouraging tensor-wide coherence with minimal additional computational overhead. Full architectural details are provided in Appendix C.

### 3.3 LATENT DIFFUSION OVER COUPLED DWI/DTI REPRESENTATIONS

Input DWIs and corresponding DTI components are coupled manifestations of the same diffusion process. Building on this principle, we introduce a generative framework operating on the latent representations (Figure 3).

Formally, let $\mathcal{Z}^{\mathbf{X}} = \{\mathbf{z}_c^{\mathbf{X}}\}_{c=1}^4$ denote the set of latent representations for the input DWI volumes, and let $\mathcal{Z}^{\mathbf{Y}} = \{\mathbf{z}_c^{\mathbf{Y}}\}_{c=1}^6$ denote the set for the target diffusion tensor components. We define the complete collection of latent variables as the union of these sets, $\mathcal{Z} = \mathcal{Z}^{\mathbf{X}} \cup \mathcal{Z}^{\mathbf{Y}}$. The forward diffusion process is defined for any individual latent component $\mathbf{z}_c \in \mathcal{Z}$. We gradually add Gaussian noise $\boldsymbol{\epsilon}$ over $T$ timesteps according to a fixed variance schedule $\beta_t$:

$$\mathbf{z}_{c,t} = \sqrt{\bar{\alpha}_t}\mathbf{z}_{c,0} + \sqrt{1 - \bar{\alpha}_t}\boldsymbol{\epsilon},$$

where $\mathbf{z}_{c,t}$ represents the noisy version of the component at timestep $t$, $\bar{\alpha}_t = \prod_{s=1}^t (1 - \beta_s)$, and $\boldsymbol{\epsilon} \sim \mathcal{N}(\mathbf{0}, \mathbf{I})$ is the noise sampled from a standard normal distribution. The generative model $\boldsymbol{\epsilon}_\theta$ is trained to reverse this process. As detailed below, our training strategy proceeds in two stages: unconditional pre-training on individual components, followed by conditional fine-tuning on the coupled field.

### 3.3.1 TENSOR AND POSITIONAL CONDITIONING

To enable the shared U-Net backbone $\boldsymbol{\epsilon}_\theta$ to process diverse inputs ranging from scalar maps to directional gradient volumes, each latent input is augmented with explicit type and position information.

(a) Latent Diffusion Model

(b) Architecture of Denoising U-Net

Figure 3: **Latent diffusion over coupled DWI/DTI representations.** (a) Both DWI ($\{\mathbf{z}_c^{\mathbf{X}}\}$) and noisy DTI ($\{\mathbf{z}_{c,t}^{\mathbf{Y}}\}$) latents are augmented with component-type and positional information. (b) The denoising U-Net architecture uses tensor-aware attention blocks to model interactions among all DWI and DTI latent components during denoising.

As shown in Figure 3a, we employ a tensor conditioning module that generates a conditioning embedding by merging learnable embeddings specific to the component (identifying the input as $B_0, D_{xx}, \dots$) with Fourier positional embeddings (Tancik et al., 2020). This combined representation is concatenated with the latent tensor before being fed into the network, ensuring the model is aware of both the physical nature and spatial context of the input component.

### 3.3.2 UNCONDITIONAL PRE-TRAINING FOR LATENT PRIOR

To stabilize training and learn a robust prior over the anatomical manifold, we first pre-train $\boldsymbol{\epsilon}_\theta$ in an unconditional setting. In this phase, the network treats each latent volume independently. The tensor-aware attention blocks designed for cross-component interaction are deactivated, and the model focuses solely on learning the distribution of valid brain structures and tensor features.

Critically, we utilize both DWI latents ($\mathcal{Z}^{\mathbf{X}}$) and DTI latents ($\mathcal{Z}^{\mathbf{Y}}$) during this stage. By training on the full set $\mathcal{Z}$, the model learns a generalized representation of the diffusion MRI domain. The

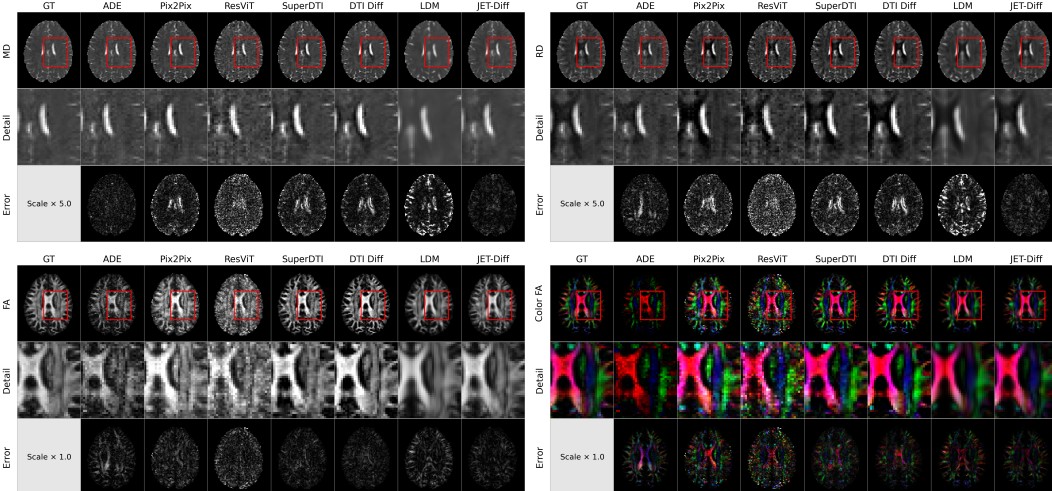

Figure 4: Qualitative comparison of DTI parameter maps (MD, RD, FA, and Color FA). JET-Diff generates reconstructions with improved anatomical fidelity and lower error. Magnified insets (red box) and error maps (scaled for visibility) highlight the detail recovered relative to the ground truth (GT) and competing methods.

objective is to predict the noise added to a single input component $\mathbf{z}_{c,t}$:

$$\mathcal{L}_{\text{pretrain}} = \mathbb{E}_{t, \mathbf{z}_{c,0}, \boldsymbol{\epsilon}} \left[ \sum_{\mathbf{z}_c \in \mathcal{Z}} \| \boldsymbol{\epsilon} - \boldsymbol{\epsilon}_\theta(\mathbf{z}_{c,t}, t) \|_2^2 \right].$$

Here, the model takes a single noisy component $\mathbf{z}_{c,t}$ as input and minimizes the reconstruction error across all available components in $\mathcal{Z}$. This allows the network to learn shared spatial features and local textures common to both DWI signals and tensor maps before modeling their complex joint dependencies.

### 3.3.3 Conditional Fine-tuning for Guided Synthesis

Following pre-training, we fine-tune the model for the target task: synthesizing the full set of DTI latents conditioned on the sparse DWI latents. In this stage, the tensor-aware attention blocks are activated, enabling the model to process the components as a coupled field. Unlike the pre-training phase, the input to the network is now the full set of noisy DTI latents $\{\mathbf{z}_{c,t}^{\mathbf{Y}}\}_{c=1}^6$, and the model is explicitly conditioned on the clean DWI latents $\mathcal{Z}^{\mathbf{X}}$. The objective function is updated to capture the joint distribution:

$$\mathcal{L}_{\text{cond}} = \mathbb{E}_{t, \mathbf{z}_0^{\mathbf{Y}}, \boldsymbol{\epsilon}} \left[ \sum_{c=1}^6 \| \boldsymbol{\epsilon}_{\mathbf{c}} - \boldsymbol{\epsilon}_{\theta, \mathbf{c}}(\{\mathbf{z}_{c,t}^{\mathbf{Y}}\}_{all}, t, \mathcal{Z}^{\mathbf{X}}) \|_2^2 \right],$$

where $\{\mathbf{z}_{c,t}^{\mathbf{Y}}\}_{all}$ denotes the set of all noisy tensor components at timestep $t$. In this formulation, tensor-aware attention allows all DWI and DTI latents to interact via a multi-axis attention mechanism. By solving this conditional denoising task, JET-Diff learns to reconstruct a geometrically consistent tensor field that faithfully reflects the anatomical information encoded in the sparse input DWIs.

## 4 Experiments

### 4.1 Setups

#### 4.1.1 Data and Preprocessing

All experiments are conducted on diffusion MRI data from the Human Connectome Project (HCP) Young Adult dataset (Van Essen et al., 2013). We utilize DWI volumes acquired at a b-value of

Table 1: Quantitative comparison of DTI parameter map synthesis. Each entry reports mean and standard deviation (mean$_{std}$) for NMSE, PSNR, and SSIM across the test set. Best and second-best results are highlighted.

| Model | MD | | | RD | | | FA | | | Color FA | | |
|---|---|---|---|---|---|---|---|---|---|---|---|---|
| | NMSE | PSNR | SSIM | NMSE | PSNR | SSIM | NMSE | PSNR | SSIM | NMSE | PSNR | SSIM |
| ADE | $0.09_{0.27}$ | $\mathbf{26.1}_{3.9}$ | $\mathbf{0.97}_{0.01}$ | $0.10_{0.31}$ | $\mathbf{27.5}_{3.7}$ | $\mathbf{0.97}_{0.01}$ | $0.30_{0.02}$ | $17.2_{0.4}$ | $0.79_{0.02}$ | $0.83_{0.03}$ | $21.4_{0.6}$ | $0.68_{0.03}$ |
| CycleGAN | $0.13_{0.01}$ | $20.0_{1.0}$ | $0.79_{0.02}$ | $0.17_{0.01}$ | $20.4_{1.4}$ | $0.78_{0.03}$ | $0.56_{0.02}$ | $14.4_{0.3}$ | $0.57_{0.04}$ | $1.16_{0.04}$ | $20.0_{0.5}$ | $0.60_{0.03}$ |
| Pix2Pix | $0.05_{0.00}$ | $24.1_{1.1}$ | $0.95_{0.00}$ | $0.06_{0.00}$ | $24.9_{1.5}$ | $0.95_{0.00}$ | $0.29_{0.02}$ | $17.3_{0.6}$ | $0.82_{0.02}$ | $1.10_{0.16}$ | $20.3_{0.7}$ | $0.72_{0.02}$ |
| ResViT | $0.07_{0.01}$ | $22.8_{1.1}$ | $0.90_{0.01}$ | $0.08_{0.01}$ | $23.6_{1.5}$ | $0.90_{0.01}$ | $0.47_{0.06}$ | $15.3_{0.5}$ | $0.73_{0.04}$ | $1.49_{0.21}$ | $18.9_{0.6}$ | $0.65_{0.02}$ |
| SuperDTI | $0.05_{0.01}$ | $24.0_{1.0}$ | $0.94_{0.01}$ | $0.06_{0.01}$ | $24.8_{1.4}$ | $0.94_{0.01}$ | $0.26_{0.02}$ | $17.8_{0.4}$ | $\underline{0.83}_{0.01}$ | $0.94_{0.07}$ | $20.9_{0.6}$ | $0.72_{0.02}$ |
| Diff-DTI | $\underline{0.05}_{0.01}$ | $\underline{24.3}_{1.0}$ | $0.95_{0.00}$ | $\underline{0.06}_{0.01}$ | $25.1_{1.3}$ | $\underline{0.95}_{0.00}$ | $\underline{0.21}_{0.02}$ | $\underline{18.7}_{0.4}$ | $\mathbf{0.89}_{0.01}$ | $\underline{0.82}_{0.06}$ | $\underline{21.5}_{0.6}$ | $\mathbf{0.85}_{0.01}$ |
| LDM | $0.11_{0.01}$ | $20.9_{1.1}$ | $0.84_{0.02}$ | $0.13_{0.01}$ | $21.6_{1.5}$ | $0.84_{0.01}$ | $0.32_{0.02}$ | $16.9_{0.5}$ | $0.69_{0.02}$ | $0.71_{0.04}$ | $22.1_{0.7}$ | $0.71_{0.03}$ |
| JET-Diff | $\mathbf{0.03}_{0.01}$ | $26.1_{1.1}$ | $\underline{0.96}_{0.01}$ | $\mathbf{0.04}_{0.01}$ | $\underline{26.6}_{1.4}$ | $\underline{0.95}_{0.01}$ | $\mathbf{0.19}_{0.01}$ | $\mathbf{19.1}_{0.5}$ | $\underline{0.83}_{0.02}$ | $\mathbf{0.62}_{0.03}$ | $\mathbf{22.7}_{0.7}$ | $\underline{0.76}_{0.03}$ |

1000 s/mm$^2$ and preprocessed with the standard HCP pipelines (Glasser et al., 2013). Ground-truth diffusion tensors are computed for each subject via a linear least-squares fit on the full set of 90 DWI directions. All DWI volumes are resampled to 2 mm isotropic resolution. The input to our model is a sparse 4-volume stack: one non-diffusion-weighted (b=0) image and the three DWI volumes whose gradient vectors are most closely aligned with the principal x, y, and z axes. The output is the complete 6-component diffusion tensor field. The full dataset of 973 subjects is partitioned into training (681), validation (97), and test (195) sets. Further details on data preparation are available in Appendix B.

Although the HCP acquisition has higher intrinsic spatial and angular resolution than typical clinical protocols, the resampled 2 mm isotropic resolution and single-shell $b = 1000$ s/mm$^2$ configuration fall within the range of many clinical DTI protocols. All experimental claims in this work are explicitly restricted to this resampled, single-shell setting on healthy young adults.

### 4.1.2 IMPLEMENTATION DETAILS

All experiments were implemented in PyTorch (Paszke et al., 2019) and conducted on a single NVIDIA A6000 GPU. Training followed a three-stage latent-diffusion pipeline. First, the Anatomical Autoencoder is pre-trained to establish a high-fidelity latent space, including an internal lightweight joint refinement block that encourages consistency across tensor components. Second, the denoising U-Net is trained as an unconditional latent diffusion model to learn a prior over the latent manifold. Third, the diffusion model is fine-tuned conditionally to synthesize tensor latents from sparse DWI latents via the coupled latent field. Detailed architectures, loss functions, and stage-specific objectives are provided in Appendix C.

### 4.1.3 COMPETING METHODS

We benchmark JET-Diff against five methods: analytic diagonal estimation (ADE), a non-learning baseline that assumes a diagonal diffusion tensor by setting off-diagonal elements to zero, and four deep learning baselines: CycleGAN (Zhu et al., 2017), Pix2Pix (Isola et al., 2017), ResViT (Dalmaz et al., 2022), SuperDTI (Li et al., 2021), Diff-DTI (Zhang et al., 2024) and a vanilla conditional latent diffusion model (Rombach et al., 2022). To ensure a fair comparison, all learning-based baselines are implemented with 3D networks and trained volumetrically on the same data splits and with identical input, except for SuperDTI and Diff-DTI, which follow their original 2D slice-based protocols. Full descriptions are available in Appendix D.

## 4.2 MAIN RESULTS

### 4.2.1 QUALITATIVE RESULTS

Figure 4 presents a qualitative comparison of the DTI parameter maps (MD, RD, FA, and Color FA) generated by JET-Diff and competing methods for a representative subject. Each row includes whole-slice views, magnified insets, and error maps relative to the ground truth. The classical approach (ADE) introduces substantial noise and structural distortions. CycleGAN fails to restore the image entirely, while Pix2Pix and ResViT produce very noisy reconstructions with limited anatomical fidelity. The standard latent diffusion baseline suppresses noise more effectively but

Table 2: Quantitative comparison of diffusion tensor components. Each entry reports the mean and standard deviation (mean$_{std}$) for PSNR, SSIM, and Log-Euclidean Metric (LEM) across the six independent tensor components ($D_{ij}$) on the test set. JET-Diff provides the most accurate and balanced reconstruction overall, with best and second-best scores highlighted.

| Model | $D_{xx}$ | | $D_{yy}$ | | $D_{zz}$ | | $D_{xy}$ | | $D_{xz}$ | | $D_{yz}$ | | LEM |
|---|---|---|---|---|---|---|---|---|---|---|---|---|---|
| | PSNR | SSIM | PSNR | SSIM | PSNR | SSIM | PSNR | SSIM | PSNR | SSIM | PSNR | SSIM | |
| ADE | $29.6_{3.0}$ | $\mathbf{0.96}_{0.01}$ | $29.4_{3.0}$ | $\mathbf{0.96}_{0.01}$ | $29.6_{3.1}$ | $\mathbf{0.96}_{0.01}$ | $24.8_{2.0}$ | $0.65_{0.05}$ | $24.4_{2.3}$ | $0.65_{0.05}$ | $24.5_{2.4}$ | $0.65_{0.05}$ | $\underline{0.51}_{0.03}$ |
| CycleGAN | $25.5_{0.7}$ | $0.81_{0.02}$ | $25.2_{0.7}$ | $0.81_{0.02}$ | $25.4_{0.7}$ | $0.81_{0.02}$ | $22.5_{2.1}$ | $0.60_{0.05}$ | $21.9_{2.3}$ | $0.59_{0.05}$ | $22.3_{2.5}$ | $0.61_{0.05}$ | $0.93_{0.04}$ |
| Pix2Pix | $29.8_{0.7}$ | $\underline{0.95}_{0.00}$ | $29.5_{0.7}$ | $\underline{0.95}_{0.00}$ | $29.6_{0.7}$ | $\underline{0.95}_{0.00}$ | $24.8_{2.1}$ | $0.73_{0.04}$ | $24.8_{2.3}$ | $\underline{0.73}_{0.04}$ | $26.9_{2.6}$ | $\mathbf{0.81}_{0.03}$ | $0.71_{0.06}$ |
| ResViT | $28.2_{0.7}$ | $0.91_{0.01}$ | $28.1_{0.7}$ | $0.91_{0.01}$ | $28.1_{0.8}$ | $0.91_{0.01}$ | $23.8_{2.1}$ | $0.67_{0.04}$ | $23.7_{2.3}$ | $0.68_{0.04}$ | $25.0_{2.5}$ | $0.72_{0.04}$ | $0.92_{0.07}$ |
| SuperDTI | $29.8_{1.0}$ | $0.94_{0.01}$ | $29.9_{1.0}$ | $0.94_{0.01}$ | $29.8_{1.0}$ | $0.94_{0.01}$ | $22.3_{2.0}$ | $0.63_{0.04}$ | $21.5_{2.3}$ | $0.63_{0.04}$ | $21.2_{2.5}$ | $0.60_{0.04}$ | $0.69_{0.05}$ |
| Diff-DTI | $\underline{30.4}_{1.5}$ | $\underline{0.95}_{0.01}$ | $\underline{30.5}_{1.4}$ | $\mathbf{0.96}_{0.01}$ | $\underline{30.6}_{1.4}$ | $\mathbf{0.96}_{0.01}$ | $22.0_{2.0}$ | $0.66_{0.04}$ | $21.4_{2.3}$ | $0.66_{0.04}$ | $20.9_{2.5}$ | $0.62_{0.04}$ | $0.64_{0.06}$ |
| LDM | $27.0_{0.8}$ | $0.87_{0.01}$ | $26.7_{0.8}$ | $0.86_{0.01}$ | $26.9_{0.8}$ | $0.86_{0.01}$ | $\mathbf{27.6}_{2.0}$ | $\underline{0.79}_{0.03}$ | $\mathbf{27.2}_{2.2}$ | $\mathbf{0.79}_{0.04}$ | $\underline{27.2}_{2.4}$ | $0.78_{0.04}$ | $0.64_{0.03}$ |
| JET-Diff | $\mathbf{31.0}_{0.7}$ | $0.95_{0.01}$ | $\mathbf{30.9}_{0.7}$ | $\underline{0.95}_{0.01}$ | $\mathbf{31.0}_{0.7}$ | $0.95_{0.01}$ | $\underline{27.5}_{2.0}$ | $\mathbf{0.80}_{0.03}$ | $\underline{27.1}_{2.2}$ | $\underline{0.79}_{0.04}$ | $\mathbf{27.3}_{2.4}$ | $\underline{0.80}_{0.04}$ | $\mathbf{0.49}_{0.03}$ |

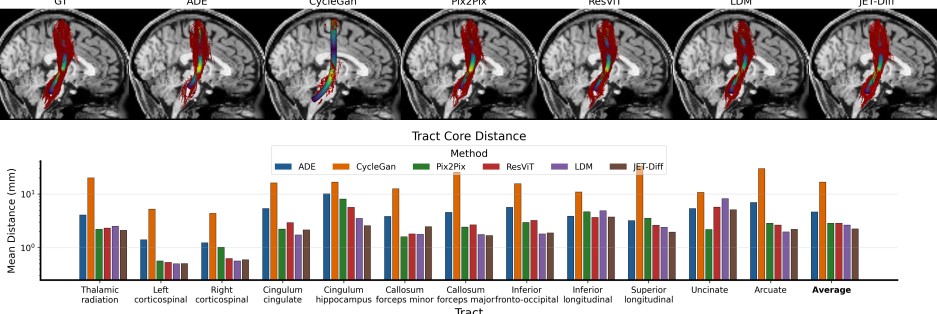

Figure 5: Tractography comparison. (Top) 3D visualization of the right corticospinal tract (CST) shows that tracts from JET-Diff tensors most closely match the ground truth. (Bottom) The mean tract core distance (mm, log scale) across 12 major white matter bundles confirms that JET-Diff yields more geometrically accurate fiber tracking compared to competing methods.

oversmooths fine structures, erasing critical white matter details. In contrast, JET-Diff achieves reconstructions that closely resemble the ground truth, suppressing noise while maintaining sharp, coherent anatomy. The error maps highlight the improvement, particularly in regions of complex fiber geometry. Figure 7 in Appendix G shows the six tensor components. Competing methods exhibit noise and blurring, especially in the off-diagonal terms ($D_{xy}, D_{xz}, D_{yz}$), which are critical yet difficult to estimate. JET-Diff yields sharper and more coherent reconstructions across all tensor elements, providing the basis for more reliable parameter maps.

### 4.2.2 QUANTITATIVE RESULTS

We quantitatively evaluated all methods using NMSE, PSNR, and SSIM (Wang et al., 2004), with results summarized in Tables 1 and 2. Detailed statistical significance tests (paired t-tests) for these metrics are provided in Appendix G (Tables 4 and 5). Further details on the geometry-aware Log-Euclidean Metric (LEM) are found in Appendix E. Because diffusion tensors reside on the SPD manifold, voxel-wise metrics such as PSNR or SSIM provide only a partial view of tensor fidelity. We therefore interpret these scores in conjunction with the geometry-aware Log-Euclidean Metric (LEM) and tractography performance. JET-Diff achieves the strongest performance across derived parameter maps, particularly for Fractional Anisotropy (FA) and Color FA, which are highly sensitive to tensor orientation and microstructural detail.

The ADE baseline highlights important limitations of conventional metrics. It achieves relatively high PSNR and SSIM scores for MD, RD, and the diagonal tensor elements, but only because it ignores the off-diagonal components. By fitting the smoother diagonal terms that dominate mean intensity, ADE secures favourable scores yet produces a degenerate tensor solution. This failure is reflected in its poor FA accuracy and inability to capture anisotropy, showing how conventional metrics can mask fundamental errors. The latent diffusion baseline illustrates a different limitation. It attains slightly higher PSNR on some off-diagonal elements than JET-Diff, but this reflects local

Table 3: Component-wise comparison of diffusion tensor reconstruction. The upper block reports autoencoder-only reconstruction, and the lower block reports full latent diffusion synthesis. Values are mean$_{\text{std}}$ for PSNR. Bold marks the best score.

| Model | Diffusion tensor component | | | | | | | DTI parameter map | | | |
|---|---|---|---|---|---|---|---|---|---|---|---|
| | $D_{xx}$ | $D_{yy}$ | $D_{zz}$ | $D_{xy}$ | $D_{xz}$ | $D_{yz}$ | LEM | MD | RD | FA | Color FA |
| *Autoencoder reconstruction* | | | | | | | | | | | |
| Ours | **35.37**$_{0.64}$ | **35.29**$_{0.68}$ | **35.35**$_{0.68}$ | **33.54**$_{1.86}$ | 33.40$_{2.09}$ | **33.80**$_{2.25}$ | **0.27**$_{0.02}$ | **33.09**$_{0.75}$ | **33.89**$_{1.04}$ | **22.27**$_{0.58}$ | **24.82**$_{0.74}$ |
| No-Joint | 35.31$_{0.64}$ | 35.24$_{0.67}$ | 35.29$_{0.67}$ | **33.54**$_{1.87}$ | **33.41**$_{2.09}$ | **33.80**$_{2.25}$ | **0.27**$_{0.02}$ | 32.96$_{0.74}$ | 33.77$_{1.03}$ | 22.23$_{0.58}$ | 24.79$_{0.73}$ |
| No-Anatomy | 31.76$_{0.57}$ | 31.68$_{0.60}$ | 31.75$_{0.60}$ | 31.07$_{1.92}$ | 30.93$_{2.15}$ | 31.38$_{2.33}$ | 0.40$_{0.02}$ | 27.11$_{0.92}$ | 27.83$_{1.29}$ | 18.93$_{0.59}$ | 23.14$_{0.75}$ |
| *Latent diffusion synthesis* | | | | | | | | | | | |
| Ours | **31.02**$_{0.67}$ | **30.83**$_{0.71}$ | **30.94**$_{0.67}$ | **27.46**$_{1.97}$ | **27.14**$_{2.19}$ | **27.32**$_{2.37}$ | **0.49**$_{0.03}$ | **26.08**$_{1.07}$ | **26.58**$_{1.37}$ | **19.12**$_{0.47}$ | **22.70**$_{0.67}$ |
| No-Joint | 30.98$_{0.68}$ | 30.80$_{0.71}$ | 30.91$_{0.68}$ | 27.41$_{1.98}$ | 27.09$_{2.20}$ | 27.30$_{2.38}$ | **0.49**$_{0.03}$ | 26.03$_{1.09}$ | 26.53$_{1.40}$ | 19.11$_{0.48}$ | **22.70**$_{0.67}$ |
| No-Pretrain | 29.17$_{0.83}$ | 28.95$_{0.86}$ | 29.06$_{0.88}$ | 25.88$_{2.01}$ | 25.63$_{2.24}$ | 25.88$_{2.39}$ | 0.60$_{0.03}$ | 24.38$_{1.30}$ | 24.46$_{1.54}$ | 17.80$_{0.36}$ | 22.11$_{0.58}$ |
| Channel | 26.87$_{0.66}$ | 27.00$_{0.73}$ | 26.79$_{0.70}$ | 22.23$_{2.07}$ | 21.94$_{2.27}$ | 22.18$_{2.46}$ | 0.89$_{0.03}$ | 22.67$_{1.11}$ | 22.44$_{1.45}$ | 15.31$_{0.17}$ | 20.44$_{0.43}$ |

voxel-wise fits rather than tensor-level coherence. JET-Diff, in contrast, achieves high accuracy on the dominant diagonal components and the lowest LEM while remaining competitive on the off-diagonals. Its coupled latent modeling yields a balanced reconstruction across all tensor elements, resulting in parameter maps such as FA and Color FA that better reflect the underlying white matter structure.

### 4.2.3 TRACTOGRAPHY COMPARISONS

To evaluate the practical utility of the reconstructed tensors, we performed whole-brain probabilistic tractography (Garyfallidis et al., 2014; Girard et al., 2014). (See Appendix F for the definition of the tract core distance metric used for evaluation). This task provides a stringent validation, as it depends on the coherence of all six tensor components and is sensitive to errors in their orientation fields. Figure 5 shows that fiber bundles generated from JET-Diff closely follow the ground truth, outperforming all competing methods both qualitatively and quantitatively. Quantitatively, JET-Diff achieves the lowest tract core distance across major white matter bundles, indicating that the reconstructed tensors are well suited for fiber tracking. Tractography provides a sensitive downstream measure of tensor coherence, and we consider it a key indicator of whether the reconstructed tensor field preserves physically meaningful orientation information beyond voxel-wise error metrics.

### 4.3 ABLATION STUDIES

We conducted ablation experiments to examine the contribution of each major component in JET-Diff. Four variants were evaluated: (1) **No-Anatomy**, which removes DWI-based anatomical conditioning and forces the autoencoder to represent both structure and tensor content within a single latent code; (2) **No-Joint**, which disables the joint refinement block in the autoencoder; (3) **No-Pretrain**, which omits unconditional latent diffusion pre-training; (4) **Channel**, which removes coupled latent modeling and processes all latent channels independently. Quantitative results for both autoencoder-only reconstruction and full latent diffusion synthesis are summarized in Table 3, with corresponding statistical significance tests reported in Table 6 in Appendix G.

### 4.3.1 ABLATION ON ANATOMICAL AUTOENCODER COMPONENTS

Removing anatomical conditioning (No-Anatomy) results in the largest degradation among autoencoder variants. While the full model achieves diagonal tensor PSNR values around 35.3–35.4 and a LEM of 0.27, No-Anatomy drops to approximately 31.7–31.8 with a higher LEM of 0.40. Parameter maps follow a similar trend, with MD PSNR decreasing from 33.1 to 27.1 and FA from 22.3 to 18.9. These reductions indicate that anatomical features supplied to the decoder are critical for disentangling spatial context from tensor-specific information.

Disabling the joint refinement block (No-Joint) produces more modest but consistent reductions. PSNR remains close to the full model (e.g., $D_{xx}$: 35.37→35.31), yet parameter maps show small declines (MD: 33.09→32.96, RD: 33.89→33.77). These results suggest that the joint refinement block improves subtle cross-component consistency without substantially altering voxel-wise reconstruction.

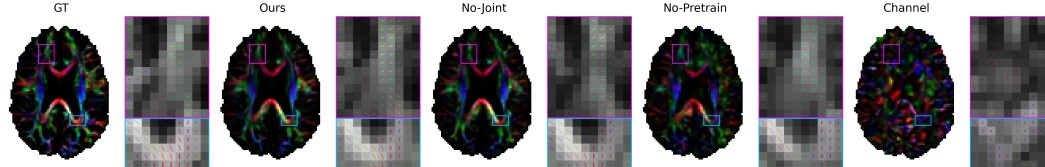

Figure 6: Ablation study on unconditional pre-training and coupled latent modeling. Results are shown for the full model, No-Pretrain, No-Joint, and Channel variants, alongside the ground truth. Removing pre-training or disrupting the coupled latent structure leads to noisier FA maps and less coherent principal eigenvector (V1) fields, as highlighted in the magnified insets.

### 4.3.2 ABLATION ON LATENT DIFFUSION COMPONENTS

The Channel variant—which removes coupled latent interactions—shows the strongest degradation during full diffusion synthesis. PSNR for off-diagonal components drops sharply (e.g., $D_{xy}$: 27.46→22.23), and LEM increases from 0.49 to 0.89. Parameter-map accuracy also falls (MD: 26.08→22.67; FA: 19.12→15.31). These results highlight the importance of modeling correlations across the six tensor components and the sparse DWI latents.

Skipping unconditional pre-training (No-Pretrain) also harms stability. Tensor PSNR decreases (e.g., $D_{xx}$: 31.02→29.17) and LEM rises from 0.49 to 0.60. Parameter maps similarly degrade (MD: 26.08→24.38). As shown in Figure 6, removing unconditional pre-training or disrupting the coupled latent structure produces noticeably noisier FA maps and less coherent principal eigenvector (V1) fields compared with the full model. The magnified insets further reveal loss of directional smoothness in the No-Pretrain and Channel variants, consistent with their higher LEM values in Table 3. This confirms that unconditional pre-training provides a stable initialization for the conditional denoising stage and improves overall geometric fidelity. Finally, we provide a detailed comparison of inference runtime and computational efficiency against baseline methods in Appendix H.

### 4.4 LIMITATIONS

This study has several limitations. All experiments are conducted on the HCP Young Adult dataset, which includes healthy young adults with uniform, high-quality acquisitions; thus, generalization to older individuals, patients with neurological conditions, or data acquired across different sites and scanners remains untested (Madden et al., 2012). In addition, our setting relies on single-shell $b = 1000$ s/mm$^2$ data resampled to 2 mm isotropic resolution with a fixed, extremely sparse four-volume DWI input. While this configuration lies within the bounds of certain clinical protocols, it differs from higher-resolution, multi-shell, or high-angular-resolution regimes used for more complex diffusion models such as fODFs or spherical deconvolution (Jeurissen et al., 2014; Alexander et al., 2019; Tournier et al., 2007). Finally, the reference tensors are generated using a single tensor fitting algorithm, and evaluating robustness across alternative fitting methods and preprocessing pipelines is an important direction for future work (Jones & Cercignani, 2010).

## 5 CONCLUSION

In this work, we introduced JET-Diff, a latent diffusion framework for reconstructing diffusion tensors from critically undersampled DWI data. JET-Diff addresses key limitations of existing methods by improving anatomical coherence across volumes and better capturing the correlations required for a coherent tensor field. The framework combines an Anatomical Autoencoder, which separates anatomical context from tensor properties to form an efficient latent space, with a diffusion process that operates on a coupled latent field of sparse DWIs and tensor components. Because the model learns a latent prior, it remains compatible with standard tensor fitting procedures while enhancing tensor-level consistency. Extensive evaluations spanning tensor components, derived DTI parameters, geometry-aware metrics such as the Log-Euclidean Metric (LEM), and downstream tractography demonstrate that, in the studied single-shell, 4-direction setting, JET-Diff improves reconstruction fidelity over competing approaches.

REPRODUCIBILITY STATEMENT

Our proposed method is designed to be reproducible. For methodology and implementation details, readers are referred to our source code, which is available in the Supplementary Material.

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

## A    DIFFUSION TENSOR MODEL DETAILS

### A.1    THE STEJSKAL-TANNER EQUATION EXPLAINED

The Stejskal-Tanner equation provides the foundational model for DTI (Stejskal & Tanner, 1965). The terms are defined as follows:

- $S_0$: The signal intensity measured in a non-diffusion-weighted acquisition (a $B_0$ image), where the diffusion-sensitizing gradients are turned off.

- $S(\mathbf{g})$: The signal intensity measured when a diffusion-sensitizing magnetic field gradient is applied along the direction of the unit vector $\mathbf{g}$.

- **b-value**: A scalar value that encapsulates the strength and duration of the diffusion gradients. A higher b-value results in greater signal attenuation for diffusing water molecules.

### A.2    TENSOR ESTIMATION AND CLINICAL CONTEXT

To solve for the six unknown components of the symmetric tensor $\mathbf{D}$, the Stejskal-Tanner equation must be sampled with at least six non-collinear gradient directions ($\mathbf{g}$). In clinical and research practice, many more directions (often 30 to 90 or more) are acquired to improve the accuracy and robustness of the tensor fit, especially in noisy data (Jones et al., 2013). This requirement leads to the primary clinical challenge of DTI: long acquisition times, which increase patient discomfort and sensitivity to motion artifacts.

### A.3    TENSOR-DERIVED METRICS

The diffusion tensor $\mathbf{D}$ is rarely interpreted directly. Instead, it is diagonalized to yield three eigenvalues ($\lambda_1 \geq \lambda_2 \geq \lambda_3$) and their corresponding eigenvectors ($\mathbf{v}_1, \mathbf{v}_2, \mathbf{v}_3$). These represent the magnitude of diffusion in three orthogonal directions and the orientation of those directions, respectively. From these, crucial microstructural metrics are calculated (Basser et al., 1994):

- **Mean Diffusivity (MD)**: The average of the eigenvalues, $MD = (\lambda_1 + \lambda_2 + \lambda_3)/3$. It measures the overall magnitude of water diffusion in a voxel, independent of directionality.

- **Radial Diffusivity (RD)**: The average of the secondary and tertiary eigenvalues, $RD = (\lambda_2 + \lambda_3)/2$. This metric quantifies diffusion perpendicular to the principal fiber direction and is often cited as a marker for myelin integrity.

- **Fractional Anisotropy (FA)**: A normalized measure of the variance of the eigenvalues, indicating the degree to which diffusion is directional. An FA of 0 implies isotropic diffusion, while an FA close to 1 implies diffusion is highly restricted to a single direction.

- **Color Fractional Anisotropy (Color FA)**: To visualize the principal fiber orientation alongside anisotropy, the FA map is modulated by the principal eigenvector $\mathbf{v}_1 = [v_{1x}, v_{1y}, v_{1z}]$. The resulting RGB image assigns colors based on direction: Red for left-right ($|v_{1x}|$), Green for anterior-posterior ($|v_{1y}|$), and Blue for superior-inferior ($|v_{1z}|$), scaled by the FA value.

These metrics are essential for the quantitative analysis of white matter integrity.

# B    DATA AND PREPROCESSING DETAILS

**Ground-Truth Tensor Generation:** The ground-truth DTI metrics for each subject were derived from the complete diffusion dataset, which included 18 b=0 volumes and 90 DWI volumes at b=1000 s/mm$^2$. Diffusion tensor fitting was performed using an ordinary linear least-squares method via the `dtifit` function in FSL (Jenkinson et al., 2012), incorporating the provided gradient nonlinearity correction files. This process yielded the full diffusion tensor, from which all ground-truth metrics, including fractional anisotropy (FA) and mean diffusivity (MD), were calculated (Glasser et al., 2013).

**Undersampled Input Selection:** The 4-volume sparse input for our model was created by selecting a specific subset of DWIs. For the b=1000 s/mm$^2$ shell, we identified the three diffusion gradient vectors most closely aligned with the standard Cartesian axes ([1, 0, 0], [0, 1, 0], and [0, 0, 1]) by minimizing the Euclidean distance. The corresponding DWI volumes were extracted, and a single b=0 s/mm$^2$ volume was prepended to form the final 4-volume input stack, $\mathcal{B}$.

# C    IMPLEMENTATION DETAILS

## C.1    JET-DIFF TRAINING PIPELINE

Our proposed method, JET-Diff, is trained in a three-stage process designed to sequentially build the model's capabilities. The first stage establishes the high-fidelity latent space via the Anatomical Autoencoder, while the final two stages train the latent diffusion model to operate within that space. We use the Adam optimizer (Kingma & Ba, 2014) for the autoencoder stage and AdamW (Loshchilov & Hutter, 2017) for the diffusion stages.

**Stage 1: Anatomical Autoencoder Pre-training.** We first train the autoencoder to learn a high-fidelity latent representation for each of the six tensor components independently. The encoder compresses each component into a latent space with 6 embedding dimensions and a codebook of 1024 entries. The architecture uses a base of 64 channels, channel multipliers of (1, 2, 4), and two residual blocks per resolution level. This phase is trained using the Adam optimizer with a learning rate of $1.0 \times 10^{-5}$ and an effective batch size of 8. The objective consists of a voxel-wise reconstruction loss and a vector-quantization commitment loss (Van Den Oord et al., 2017).

**Stage 2: Decoupled Joint Refinement.** After pre-training, we freeze the autoencoder weights and fine-tune a new joint decoder to enforce consistency across all six tensor components. All weights are frozen except for a new joint fusion block and the final output convolution layers of the decoder. Optimization uses the Adam optimizer with a learning rate of $1.0 \times 10^{-4}$ and an effective batch size of 16.

**Stage 3: Unconditional Latent Diffusion Pre-training.** To provide a strong initialization for the generative model, we pre-train the denoising U-Net to model the distribution of the latent tensors in an unconditional setting. The backbone has 256 base channels, channel multipliers of (1, 2, 4), two residual blocks per scale, and self-attention at multiple resolutions. The diffusion process uses a linear beta schedule (Ho et al., 2020) over 1000 steps. In this phase, the model is trained to denoise the latent tensors without explicit conditioning, learning a robust prior over the latent manifold. The model is trained with the AdamW optimizer with a learning rate of $1.0 \times 10^{-6}$ and a batch size of 8.

**Stage 4: Conditional Latent Diffusion Fine-tuning.** The model is then fine-tuned for the primary conditional synthesis task, initialized from the checkpoint of the unconditional pre-training phase. The forward diffusion process continues to apply Gaussian noise to the coupled latent field. The denoising U-Net is now conditioned on the clean DWI latents, which are concatenated to the noisy latent tensor and modulate the tensor-aware attention blocks. The model is trained to predict the noise for all channels, guided by the DWI condition, using AdamW with a learning rate of $1.0 \times 10^{-6}$ and an effective batch size of 8.

## C.2    JOINT MLP BLOCK ARCHITECTURE

The joint refinement block added at the end of the decoder operates voxel-wise across all six tensor components. Its architecture is as follows:

- **Input dimension**: 1536 (= 256 channels × 6 tensor components)
- **Hidden dimension**: 768 (GELU activation)
- **Output dimension**: 1536
- **Total parameters**: 2,361,600

This block contributes only a small fraction of the decoder's total parameters and is designed to promote cross-component consistency with minimal overhead.

### C.3 COMPARISON: COUPLED LATENT ATTENTION VS. STANDARD CROSS-ATTENTION

To clarify the distinction between the attention mechanism employed in our denoising U-Net and the standard Cross-Attention mechanism commonly used in latent diffusion models, we provide a detailed formulation comparison.

#### C.3.1 STANDARD CROSS-ATTENTION

In a standard conditional diffusion model, the generative backbone (U-Net) synthesizes the target $\mathbf{z}$ conditioned on an input $\mathbf{c}$. The Cross-Attention mechanism facilitates this by attending to the condition $\mathbf{c}$ using queries derived from the intermediate representation of the target $\mathbf{z}$. Formally:

$$\text{Attention}(Q, K, V) = \text{softmax}\left(\frac{QK^T}{\sqrt{d}}\right) V \tag{1}$$

where $Q = W_Q \cdot \varphi(\mathbf{z}_t)$, $K = W_K \cdot \psi(\mathbf{c})$, and $V = W_V \cdot \psi(\mathbf{c})$. Here, the information flow is asymmetric and unidirectional ($\mathbf{c} \to \mathbf{z}$). Crucially, this mechanism treats the channels of $\mathbf{z}$ independently in terms of the attention logic; it does not explicitly model the correlation between different output channels (e.g., tensor components) beyond what is captured by convolutional receptive fields.

#### C.3.2 COUPLED LATENT ATTENTION

In the context of DTI reconstruction, valid tensor synthesis requires strict coherence between the six tensor components ($\mathbf{Y}_c$). Standard cross-attention is insufficient because it does not explicitly enforce this inter-component consistency.

Our proposed framework addresses this by employing a **Coupled Latent Attention** mechanism that models the *joint distribution* of the input DWIs ($\mathbf{X}$) and output DTIs ($\mathbf{Y}$). We construct a unified sequence $S$ that concatenates the tokens of both the conditioning DWI latents and the noisy DTI latents along the component dimension. This mechanism is implemented as a modified Self-Attention over this joint sequence:

$$Q, K, V = W_{Q,K,V} \cdot [\mathbf{z}_t^{\mathbf{Y}}; \mathbf{z}^{\mathbf{X}}] \tag{2}$$

This formulation introduces three key contributions:

1. **Symmetric Interaction:** Unlike cross-attention, our approach enables "all-to-all" communication. This allows specific DTI components to attend to other DTI components (e.g., $D_{xy}$ attending to $D_{xx}$ and $D_{yy}$), explicitly enforcing the physical constraints of the diffusion tensor via the attention map.

2. **Directional Geometry Awareness:** As described in Section 3.3.1, we incorporate learnable direction embeddings ($\mathbf{E}_{dir}$) directly into the attention calculation:

$$\text{Sim}(Q, K) = \frac{QK^T}{\sqrt{d}} + \mathbf{E}_{pos} + \mathbf{E}_{dir} \tag{3}$$

This explicitly informs the network of the geometric relationship between the signal gradients (in DWIs) and the tensor orientation components (in DTIs), a feature absent in standard spatial cross-attention.

### C.4 TRAINING SCHEDULES FOR ABLATION VARIANTS

For completeness, we report the exact number of training iterations used for each ablation model, ensuring that performance differences are attributable to architectural factors rather than computational budget:

- **JET-Diff (full model)**: 75k steps (autoencoder pre-training) + 5k (joint refinement) + 25k (unconditional diffusion) + 25k (conditional diffusion)
- **No-Anatomy**: 120k autoencoder-only steps
- **No-Pretrain**: 100k diffusion steps
- **Channel variant**: 20k diffusion steps

## D COMPETING METHODS DETAILS

Unless otherwise noted, all learnable baselines are implemented as 3D networks and trained on full 3D volumes to ensure a rigorous comparison with our volumetric approach.

- **ADE (Analytic Diagonal Estimation):** A non-learning baseline that assumes a diagonal diffusion tensor. The diagonal components ($D_{xx}$, $D_{yy}$, $D_{zz}$) are computed from the log-linearized Stejskal-Tanner equation using the most aligned gradients. Off-diagonal elements are strictly set to zero, and the tensor is projected onto the Symmetric Positive Definite (SPD) manifold.
- **SuperDTI (Li et al., 2021):** Implemented using a 2D U-Net-style encoder–decoder CNN with residual learning. Following the original protocol, the network is trained to learn the non-linear mapping from uniformly sampled sparse DWI signals directly to FA, MD, and the primary eigenvectors (or color maps) using an $L_2$ regression loss. We utilized the architecture specified in the official paper, adapted for 2D based processing.
- **Diff-DTI (Zhang et al., 2024):** A conditional score-based diffusion model. The method operates on 2D slices to directly synthesize DTI parametric maps (e.g., FA, MD, Color FA) from a few conditional DWIs. The model uses a novel U-Net backbone enhanced by a Feature Enhancement Fusion Mechanism (FEFM), which integrates a Transformer-based auxiliary path to preserve fine structural details, and is trained with a score-matching objective.
- **CycleGAN (Zhu et al., 2017):** The architecture consists of two 3D U-Net generators and two 3D PatchGAN discriminators, trained with an adversarial loss and an L1 cycle-consistency loss ($\lambda = 10$).
- **Pix2Pix (Isola et al., 2017):** The generator is a 3D U-Net, and the discriminator is a 3D PatchGAN. The training objective is a sum of a vanilla GAN loss and an L1 reconstruction loss ($\lambda_{L1} = 100$).
- **ResViT (Dalmaz et al., 2022):** A hybrid architecture combining a 3D ResNet-style backbone with interleaved Vision Transformer blocks to capture long-range dependencies, trained with a composite L1 and adversarial loss.
- **LDM (Rombach et al., 2022):** A standard Latent Diffusion Model serves as a comparative baseline. We employ a conventional latent autoencoder *identical to the No-Anatomy ablation*, thereby reproducing the canonical LDM setup without DWI-aided decoding or anatomical feature injection. The diffusion U-Net is conditioned solely through channel-wise concatenation of the DWI latent and the noisy tensor latent, following the design principles of SR3 (Saharia et al., 2022b) and Palette (Saharia et al., 2022a). This configuration reflects the standard conditional LDM mechanism and stands in contrast to the coupled latent field and tensor-aware joint-encoding used in JET-Diff.

### D.1 RECONSTRUCTING DIFFUSION TENSORS FROM PARAMETER MAPS

SuperDTI and Diff-DTI do not directly output the full diffusion tensor, but instead predict diffusion parameter maps such as mean diffusivity (MD), radial diffusivity (RD), fractional anisotropy (FA),

and Color FA (RGB-encoded FA). To enable tensor-domain metrics and tractography, we reconstruct an approximate diffusion tensor $D \in \mathbb{R}^{3 \times 3}$ in each voxel from these quantities under a cylindrically symmetric model.

We assume a single-shell acquisition and impose

$$\lambda_2 = \lambda_3 = \mathrm{RD},$$

so that the tensor has one principal eigenvalue $\lambda_1 = \mathrm{AD}$ (axial diffusivity) and two identical secondary eigenvalues $\lambda_2 = \lambda_3 = \mathrm{RD}$. Using the standard relation

$$\mathrm{MD} = \frac{\lambda_1 + \lambda_2 + \lambda_3}{3} = \frac{\mathrm{AD} + 2\,\mathrm{RD}}{3},$$

we recover axial diffusivity as

$$\mathrm{AD} = 3\,\mathrm{MD} - 2\,\mathrm{RD}.$$

In practice, we clip negative values of AD to zero to avoid clearly unphysical eigenvalues.

The principal eigenvector $e_1 \in \mathbb{R}^3$ is estimated from the Color FA image. Let $\mathbf{c} = (c_x, c_y, c_z)$ denote the RGB-FA vector at a voxel; we normalize it to obtain

$$e_1 = \frac{\mathbf{c}}{\|\mathbf{c}\|_2 + \varepsilon},$$

with a small $\varepsilon > 0$ for numerical stability. To suppress unreliable orientations in nearly isotropic voxels, we set $e_1 = 0$ whenever $\mathrm{FA} < 0.05$.

Under these assumptions, the reconstructed diffusion tensor is given by the rank-1 update

$$D = \mathrm{RD}\,I_3 + (\mathrm{AD} - \mathrm{RD})\,e_1 e_1^{\top},$$

where $I_3$ is the $3 \times 3$ identity matrix. Writing $e_1 = (e_x, e_y, e_z)$ and $\Delta = \mathrm{AD} - \mathrm{RD}$, the six independent components are

$$D_{xx} = \mathrm{RD} + \Delta\,e_x^2,$$
$$D_{yy} = \mathrm{RD} + \Delta\,e_y^2,$$
$$D_{zz} = \mathrm{RD} + \Delta\,e_z^2,$$
$$D_{xy} = \Delta\,e_x e_y,$$
$$D_{xz} = \Delta\,e_x e_z,$$
$$D_{yz} = \Delta\,e_y e_z.$$

We then store $(D_{xx}, D_{yy}, D_{zz}, D_{xy}, D_{xz}, D_{yz})$ as a 4D volume and use this reconstructed tensor field for all tensor-domain metrics (e.g., LEM) and tractography-based evaluations reported for SuperDTI and Diff-DTI.

# E    DIFFUSION TENSOR METRICS AND MANIFOLD GEOMETRY

To provide a geometrically rigorous assessment of the reconstructed diffusion tensors, we employ the Log-Euclidean Metric (LEM) in addition to conventional image-based metrics (PSNR, SSIM). This addresses the fundamental limitation that the space of Diffusion Tensor Imaging (DTI) tensors does not conform to Euclidean geometry.

## E.1    LIMITATION OF EUCLIDEAN METRICS

A diffusion tensor $\mathbf{D}$ is represented by a $3 \times 3$ Symmetric Positive Definite (SPD) matrix. The space of all such matrices, $\mathcal{S}_{++}^3$, forms a non-linear Riemannian manifold rather than a flat Euclidean space. Applying standard Euclidean metrics (e.g., $L_2$ norm, PSNR, SSIM) to the six unique tensor components treats them as a simple 6D vector. This approach ignores:

- Physical Plausibility: It does not enforce the positive definiteness of the tensor (i.e., non-negative eigenvalues), which is a core physical constraint of diffusion.
- Geodesic Distance: The resulting distance metric does not correspond to the shortest path (geodesic) between two tensors on the manifold, leading to potentially inaccurate assessment in regions of high anisotropy or complex fiber geometry.

### E.2 Log-Euclidean Metric (LEM)

The Log-Euclidean Metric (LEM) (Arsigny et al., 2006) provides an effective and computationally tractable distance metric for SPD matrices. It utilizes the matrix logarithm ($\log$) to map the curved SPD manifold ($\mathcal{S}_{++}^3$) to a flat vector space (the space of symmetric matrices, $\mathcal{S}^3$), where standard Euclidean operations become valid.

The Log-Euclidean distance between two tensors, $\mathbf{D}_1$ and $\mathbf{D}_2$, is defined as the Frobenius norm of the difference between their matrix logarithms:

$$d_{LE}(\mathbf{D}_1, \mathbf{D}_2) = \|\log(\mathbf{D}_1) - \log(\mathbf{D}_2)\|_F$$

Here, $\log(\mathbf{D})$ is computed via eigendecomposition ($\mathbf{D} = \mathbf{V}\mathbf{\Lambda}\mathbf{V}^T$, where $\mathbf{\Lambda}$ contains the eigenvalues $\lambda_i$), such that $\log(\mathbf{D}) = \mathbf{V}\log(\mathbf{\Lambda})\mathbf{V}^T$. In our implementation, we ensure numerical stability by clamping all eigenvalues to a minimum positive value before applying the logarithm. By minimizing this metric, we enforce that the reconstructed tensor not only matches the ground truth in component values but also adheres to the appropriate underlying tensor geometry, supporting physically plausible downstream analyses like tractography.

## F  Tractography Evaluation: Tract Core Distance

To assess how tensor reconstruction quality affects downstream fiber tracking (Section 4), we measure the geometric discrepancy between white matter bundles obtained from reconstructed tensors and those from the reference tensors using a *tract core distance* (Garyfallidis et al., 2014; Girard et al., 2014).

For each subject, whole-brain probabilistic tractography is run on the reference and on each reconstructed tensor field with identical MRtrix3 seeding and tracking parameters (Tournier et al., 2019). Major bundles (e.g., corticospinal tract, cingulum, corpus callosum segments) are segmented from the whole-brain tractograms using TractSeg-derived bundle masks. For each bundle, we build a smooth 3D *core* trajectory that summarizes its geometry: all streamlines are resampled to a fixed number of points in world coordinates, and a low-degree polynomial is fitted to the cross-sectional centroid positions along the main bundle direction. This yields a core trajectory for the ground-truth tensors, $\gamma^{\mathrm{GT}}$, and for each reconstruction, $\gamma^{\mathrm{rec}}$.

The tract core distance for a bundle is defined as the average nearest-point distance from the reconstructed core to the reference core:

$$d_{\mathrm{core}}(\gamma^{\mathrm{GT}}, \gamma^{\mathrm{rec}}) = \frac{1}{M} \sum_{m=1}^{M} \min_{1 \le k \le K} \left\| \gamma^{\mathrm{GT}}(k) - \gamma^{\mathrm{rec}}(m) \right\|_2,$$

where $\{\gamma^{\mathrm{GT}}(k)\}_{k=1}^{K}$ and $\{\gamma^{\mathrm{rec}}(m)\}_{m=1}^{M}$ are the sampled points along the two core trajectories in physical space (mm). For 2 mm isotropic data we treat sub-voxel discrepancies as negligible by thresholding very small distances. Lower values of $d_{\mathrm{core}}$ indicate that the reconstructed tensor yields a bundle whose central trajectory closely matches that of the ground truth.

## G  Additional Quantitative Results

This section contains supplementary quantitative results, including qualitative visualizations of the tensor components (Figure 7), detailed statistical significance tests (Tables 4, 5, 6), and inference runtime comparisons (Table 8).

## H  Inference Runtime Analysis

Table 8 summarizes the end-to-end inference time for a full 2 mm whole-brain volume on a single NVIDIA A6000. JET-Diff incurs additional computational cost relative to a LDM due to coupled latent attention and tensor-aware refinement, while achieving markedly higher tensor consistency. Importantly, JET-Diff remains substantially faster than 2D slice-based diffusion models such as Diff-DTI, which require processing each slice independently and accumulate significant overhead.

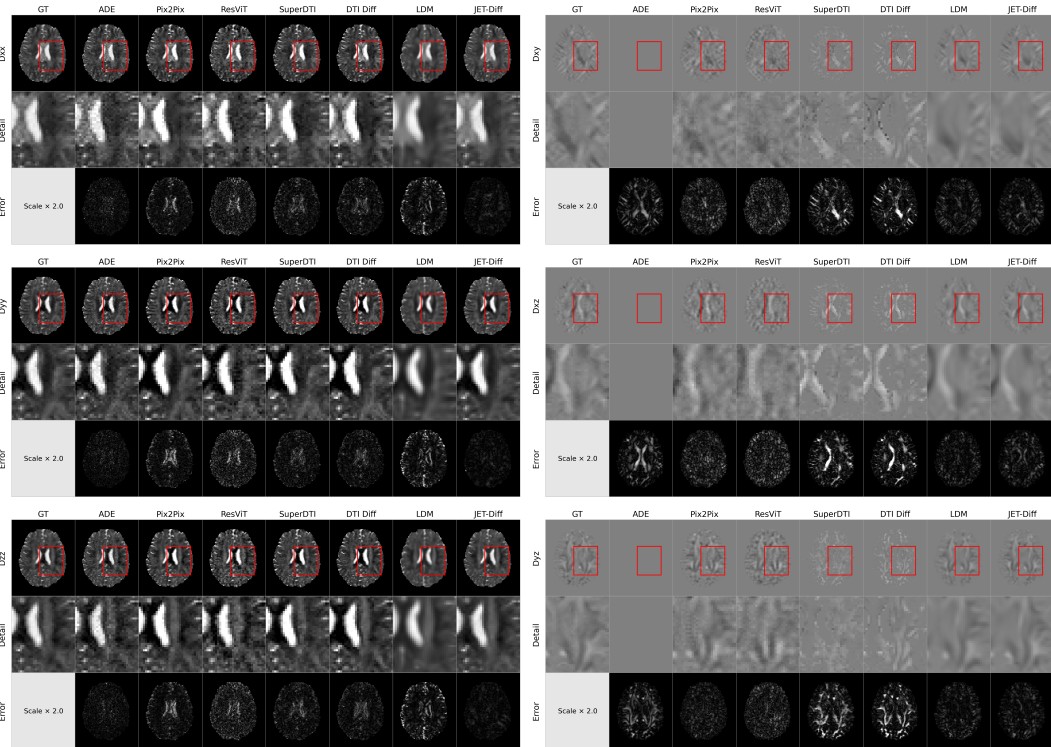

Figure 7: Qualitative comparison of diffusion tensor components. Visualization of the six individual tensor components for the same subject shown in Figure 4. JET-Diff provides a faithful reconstruction across all components, with reduced noise and artifacts, particularly in the off-diagonal elements.

Table 4: Paired t-test results (JET-Diff vs baselines) for each **Metric** (MD, FA, RD) under each **Measure** (NMSE, PSNR, SSIM). Legend: *** $p < 10^{-3}$, ** $p < 10^{-2}$, * $p < 0.05$, (n.s.) $p \geq 0.05$.

| Metric | Measure | ADE | CycleGAN | Pix2Pix | ResViT | Diff-DTI | SuperDTI | LDM |
|--------|---------|-----|----------|---------|--------|----------|----------|-----|
| MD | NMSE | 0.005** | *** | *** | *** | *** | *** | *** |
| FA | NMSE | *** | *** | *** | *** | *** | *** | *** |
| RD | NMSE | 0.016* | *** | *** | *** | *** | *** | *** |
| MD | PSNR | 0.842 (n.s.) | *** | *** | *** | *** | *** | *** |
| FA | PSNR | *** | *** | *** | *** | *** | *** | *** |
| RD | PSNR | 0.008** | *** | *** | *** | *** | *** | *** |
| MD | SSIM | *** | *** | *** | *** | *** | *** | *** |
| FA | SSIM | *** | *** | *** | *** | *** | 0.792 (n.s.) | *** |
| RD | SSIM | *** | *** | *** | *** | *** | *** | *** |

# I   USE OF AI TOOLS IN MANUSCRIPT PREPARATION

The authors utilized Google's Gemini Pro to improve the grammar and readability of this manuscript. All content generated by this tool was critically reviewed, fact-checked, and substantially revised by the authors to ensure accuracy and originality. The final responsibility for the content of this paper rests solely with the authors.

Table 5: Paired t-test results (JET-Diff vs baselines) for **Metric = LEM or Component**, under **Measure = PSNR, SSIM**. Legend: *** $p < 10^{-3}$, ** $p < 10^{-2}$, * $p < 0.05$, (n.s.) $p \geq 0.05$.

| Metric | Component / Measure | ADE | CycleGAN | Pix2Pix | ResViT | Diff-DTI | SuperDTI | LDM |
|---|---|---|---|---|---|---|---|---|
| LEM | – | *** | *** | *** | *** | *** | *** | *** |
| PSNR | Dxx | *** | *** | *** | *** | *** | *** | *** |
| PSNR | Dxy | *** | *** | *** | *** | *** | *** | *** |
| PSNR | Dxz | *** | *** | *** | *** | *** | *** | *** |
| PSNR | Dyy | *** | *** | *** | *** | *** | *** | *** |
| PSNR | Dyz | *** | *** | *** | *** | *** | *** | *** |
| PSNR | Dzz | *** | *** | *** | *** | *** | *** | *** |
| SSIM | Dxx | *** | *** | *** | *** | 0.006** | *** | *** |
| SSIM | Dxy | *** | *** | *** | *** | *** | *** | 0.021* |
| SSIM | Dxz | *** | *** | *** | *** | *** | *** | *** |
| SSIM | Dyy | *** | *** | *** | *** | *** | *** | *** |
| SSIM | Dyz | *** | *** | *** | *** | *** | *** | *** |
| SSIM | Dzz | *** | *** | *** | *** | *** | *** | *** |

Table 6: Paired t-test results for ablation study (Ours vs ablated variants), organized by **Metric** (LEM, MD–PSNR, FA–PSNR, RD–PSNR, Component PSNR). Legend: *** $p < 10^{-3}$, ** $p < 10^{-2}$, * $p < 0.05$, (n.s.) $p \geq 0.05$.

| Stage | Metric / Component | No-Joint | No-Anatomy | Channel | No-Pretrain |
|---|---|---|---|---|---|
| *Autoencoder reconstruction* | LEM | *** | *** | – | – |
| | MD–PSNR | *** | *** | – | – |
| | FA–PSNR | *** | *** | – | – |
| | RD–PSNR | *** | *** | – | – |
| | Dxx | *** | *** | – | – |
| | Dxy | 0.183 (n.s.) | *** | – | – |
| | Dxz | 0.003** | *** | – | – |
| | Dyy | *** | *** | – | – |
| | Dyz | 0.861 (n.s.) | *** | – | – |
| | Dzz | *** | *** | – | – |
| *Latent diffusion synthesis* | LEM | *** | – | *** | *** |
| | MD–PSNR | 0.012* | – | *** | *** |
| | FA–PSNR | 0.119 (n.s.) | – | *** | *** |
| | RD–PSNR | 0.007** | – | *** | *** |
| | Dxx | 0.003** | – | *** | *** |
| | Dxy | *** | – | *** | *** |
| | Dxz | *** | – | *** | *** |
| | Dyy | 0.036* | – | *** | *** |
| | Dyz | 0.144 (n.s.) | – | *** | *** |
| | Dzz | 0.031* | – | *** | *** |

Table 7: Inference runtime on a single NVIDIA A6000 (48GB). JET-Diff is slower than vanilla LDM due to coupled latent attention and tensor-aware refinement, yet remains substantially faster than score-based methods.

| Model | Inference Time (s) |
|---|---|
| SuperDTI | 10.93 |
| LDM | 23.58 |
| JET-Diff (Ours) | 122.96 |
| Diff-DTI | 593.22 |

Table 8: Inference runtime on a single NVIDIA A6000 (48GB). JET-Diff is slower than vanilla LDM due to coupled latent attention and tensor-aware refinement, yet remains substantially faster than score-based methods.

| Model | Inference Time (s) |
|---|---|
| SuperDTI | 10.93 |
| LDM | 23.58 |
| JET-Diff (Ours) | 122.96 |
| Diff-DTI | 593.22 |

