# OpenReview forum: "JET-Diff: Joint-Encoding Tensor Diffusion Model for Accurate DTI Reconstruction from Sparse DWIs"
_ICLR.cc/2026/Conference — Submitted to ICLR 2026_

### Official Review · Reviewer_8hHQ · 2025-10-25

**Soundness:** 2
**Presentation:** 2
**Contribution:** 2
**Rating:** 2
**Confidence:** 4

**Summary:**

This paper proposes a diffusion-model-based approach for DTI reconstruction from undersampled DWI data. The method trains a latent diffusion model to synthesize DTI conditioned on DWI measurements. The main novelties are: 1) a new autoencoder architecture for the latent space, which uses multi-scale features from DWI to improve the decoding of DTI images from the latent space, as well as a "Joint MLP block" to capture correlations among tensor components; 2) an additional pretraining stage to learn the unconditional joint distribution of DTI and DWI, before training the conditional diffusion model for DTI synthesis. Experiments were conducted on the public human connectome dataset and includes comparisons to existing methods as well as ablation studies on each proposed component.

**Strengths:**

- The paper identifies important problems in existing approaches for DTI reconstruction, e.g., Diff-DTI -- the anatomical inconsistency due to using 2D models and the physical implausibility caused by directly estimating DTI-derived parameter maps without reconstructing the tensor image itself.

**Weaknesses:**

- The rationales for the proposed approaches can be better explained -- some components seem redundant and could be replaced with simpler alternatives. Empirical evidence that supports their benefits can also be improved.
- Most technical improvements appear specific for the DTI problem, with limited generalizability.
- Naming of some method components can be misleading. Some standard techniques, e.g., cross-attention, are given a new name, e.g., "Joint-Encoding Tensor Attention". The contributions and novelties can be better clarified.
- Several notations, abbreviations, and technical terms are not clearly defined, making the paper relatively hard to follow.

**Questions:**

- On the proposed pretraining step
  - What is motivation for learning the joint distribution of DTI and DWI? Why will it benefit the reconstruction of DTI from DWI? Given that the problem is not to synthesize entirely new pairs of DTI-DWI data, but only to estimate DTI given DWI inputs, learning this joint distribution seems an overkill and unnecessary.
  - Empirical evidence supporting the benefit of this pretraining step is lacking. Currently, only visual comparison on one test example is provided between the models trained with and without this pretraining step (Section 4.3.3). No numerical comparisons (e.g., PSNR over the entire test set) were presented.
  - In the existing visual comparison, were the two models trained for the same number of epochs? If not, this may lead to an unfair comparison.

- On the proposed latent diffusion model and autoencoder
  - The autoencoder is first trained on each of the six tensor components independently. Then, to model inter-correlation among tensor components, the authors proposed to add a "Joint MLP block" at the end of the decoder. Why is such a two-stage approach employed, in favor of a more straightforward approach that applies the autoencoder to all tensor components simultaneously, by viewing them as a six-channel image? Have the authors compared these two approaches?
  - Similarly, why is the diffusion model applied independently to each tensor component (line 218), rather than to the entire tensor image directly, by treating it as a six-channel image? Have the authors compared these two approaches?
  - Can the authors clarify the difference between "Joint-Encoding Tensor Attention" and the standard Cross-Attention mechanism used in diffusion models for conditional image synthesis?
  - Can the authors clarify the meaning of their name "Multi-Tensor diffusion model"? Based on my understanding, this name suggests that the diffusion model is implemented for both DWI and DTI, hence "Multi-Tensor". However, DWI is not a tensor image -- it's a sequence of measurements obtained at different gradient directions.

- No comparison is conducted with existing DTI reconstruction methods, e.g., Diff-DTI, whose limitations motivated this work.

- Questions on notations:
  - Equation in line 82: The symbols are undefined.
  - Line 220: $\epsilon_c$ is not defined.
  - What does $z_{c,t}$ represent? Although $z_{c,t}^X$ and $z_{c,t}^Y$ are defined, the notation without superscript is not. Based on Section 3.3.3, it seems that $z_{c,t}$ is an abbreviation of the DTI latents $z_{c,t}^Y$. If so, consider unifying the notation.
  - In the defition of pretraining loss (line 252), why is there {} around $z_{c,t}$? If it denotes the set of $z_{c,t}^Y$ for all $c$, why is there still a sum over $c$ in the equation?
  - Consider adding definitions for MD, RD, FA and Color FA for readers not familiar with DTI.
  - Consider defining mean tract core distance.

Other comments:
- The comparison with CycleGAN may not be meaningful, since CycleGAN is developed for unpaired data. This work assumes that paired data is available for training, and, I assume, all other methods are trained on paired data, which makes this an unfair comparison.
- In the ablation experiment on autoencoder, what is the design of the "standard autoencoder"? Was it trained only on DTI? If so, how is the subsequent diffusion model adapted, which takes both the latent representations of DWI and DTI as input?

---

> ### Author Response · Authors · 2025-11-28
> **Response to Reviewer 8hHQ (Part 1)**
>
> We thank the reviewer for their meticulous and constructive feedback. The points raised concerning the motivation for unconditional pre-training, architectural choices, naming conventions, and notational clarity have led to substantial improvements in the revised manuscript. We address each issue systematically below.
>
> ## 1. Motivation and Empirical Evidence for Unconditional Pre-training
>
> ### 1.1 Rationale for the Unconditional Latent Prior
>
> We acknowledge the request for a clearer justification of the unconditional pre-training step in a conditional generation task (DTI given DWI). As clarified in the revised Sections 3.1 and 3.3.2, the core challenge lies in the ill-posed nature of reconstructing a full six-component diffusion tensor ($\mathbf{D}$) from an extremely sparse four-volume input ($\mathbf{X}_{\text{4-vol}}$).
>
> The unconditional stage (Equation 3) trains the denoising network $\epsilon_\theta$ to model the joint distribution over the coupled latent space $\mathbf{Z} = \mathbf{Z}_X \cup \mathbf{Z}_Y$ (DWI and DTI latents). This step establishes what constitutes a valid brain structure and a consistent diffusion tensor field before introducing the conditional constraint.
>
> The revised manuscript includes a rigorous comparison with the No-Pretrain ablation, which removes the unconditional stage and trains the conditional diffusion model for an equal or greater number of conditional updates (100k steps) than the full JET-Diff model (50k diffusion steps total).
>
> The results consistently show that the unconditional-to-conditional staging is essential, not merely a consequence of the total optimization budget.
>
> | Metric (Lower is Better) | JET-Diff (Full)  | No-Pretrain Ablation | Degradation     |
> | ------------------------ | ---------------- | -------------------- | --------------- |
> | LEM (Geometry)           | $0.49 \pm 0.03$  | $0.60 \pm 0.03$      | 22.4% worse     |
> | $D_{xx}$ PSNR (dB)       | $31.02 \pm 0.67$ | $29.17 \pm 0.83$     | $1.85$ dB lower |
> | FA PSNR (dB)             | $19.12 \pm 0.47$ | $17.80 \pm 0.36$     | $1.32$ dB lower |
>
> These differences achieve statistical significance ($p < 10^{-3}$) (Appendix G, Table 6). Qualitatively, the No-Pretrain variant exhibits visibly noisier FA maps and less coherent principal eigenvector fields ($\mathbf{v}_1$) (Figure 6). This demonstrates that the learned unconditional prior significantly enhances the geometric fidelity and structural coherence of the final tensor reconstruction.
>
> ## 2. Latent Diffusion and Autoencoder Design
>
> ### 2.1 Justification for the Two-Stage Anatomical Autoencoder
>
> Our Anatomical Autoencoder (AAE) (Figure 2) addresses the entanglement problem common in VAEs, whereby the latent code is forced to encode both anatomical context ("where") and tensor attributes ("what") into a single vector, a limitation analyzed in the $\beta$-VAE literature.
>
> **AAE design:** The DTI latent code $Z_Y$ focuses on tensor content, whereas the DWI conditioner $C(\mathbf{X})$ provides anatomical context ($\Phi_X$). The decoder fuses $Z_Y$ and $\Phi_X$ using cross-attention.
>
> **No-Anatomy ablation:** This variant approximates the "single six-channel AE" by removing the DWI conditioner and training exclusively on DTI.
>
> The autoencoder reconstruction results (Table 3, upper block) confirm the necessity of the AAE: the No-Anatomy variant substantially degrades MD PSNR ($33.09 \to 27.11$ dB) and LEM ($0.27 \to 0.40$). This strong empirical evidence supports the multi-stage AAE design as a key contribution to disentangled representation learning in multi-field medical imaging.
>
> ### 2.2 Joint Denoising vs. Per-Component Diffusion
>
> Joint processing is achieved through the coupled latent attention mechanism (Section 3.3.3), wherein a self-attention layer operates over the concatenated sequence $S = [Z_t^Y; Z_X]$, enabling cross-component interactions (e.g., $D_{xy}$ attending to $D_{xx}$ and $D_{yy}$). This variant removes the structured coupling and processes channels independently within the U-Net.
>
> The Channel ablation yields the largest performance decline among diffusion variants (Table 3, lower block): LEM increases from $0.49 \to 0.89$, and $D_{xy}$ PSNR drops markedly from $27.46 \to 22.23$ dB. These findings confirm that explicit cross-component coupling within the diffusion U-Net is crucial for reconstructing geometrically consistent off-diagonal tensor elements.

---

> ### Author Response · Authors · 2025-11-28
> **Response to Reviewer 8hHQ (Part 2)**
>
> ## 3. Naming and Perceived Generality
>
> ### 3.1 Naming of Attention Mechanism
>
> We concur that the original term "JET Attention" could be misinterpreted. We have replaced it with "Coupled Latent Attention" (Section 3.3.1; Appendix C.3) and now explicitly describe it as a self-attention mechanism applied to the concatenated latent sequence $S = [z_t^Y; z_X]$, distinguishing it from the unidirectional cross-attention used in conditional latent diffusion models.
>
> ### 3.2 "Multi-Tensor" Terminology
>
> We have revised the terminology to avoid referring to DWI inputs as "multi-tensor." We now use the phrase "latent diffusion over coupled DWI/DTI representations" (Section 3.3) to clarify that the model processes multiple latent fields (four DWI volumes and six DTI components) without implying that DWI itself is a tensor.
>
> ### 3.3 Generalizability
>
> We have strengthened Section 4.4 (Limitations) to emphasize that our claims are restricted to single-shell, four-direction DTI reconstruction in healthy HCP data. We refrain from suggesting generalizability to higher-order models (fODFs) or non-DTI modalities.
>
> ## 4. Baselines and Evaluation Metrics
>
> ### 4.1 Inclusion of Diff-DTI and SuperDTI
>
> We have implemented and included Diff-DTI and SuperDTI as DTI-specific baselines, retraining them on our four-volume protocol for fair comparison (Sections 2.3, 4.1.3, Tables 1–2). JET-Diff consistently outperforms both baselines on geometric and downstream metrics (LEM, tract core distance), establishing state-of-the-art performance in this sparse regime.
>
> ### 4.2 Notation and Metric Definitions
>
> We have unified notation across the manuscript (Section 5.1), providing:
>
> * Clear definitions for input ($\mathbf{X}$), output ($\mathbf{Y}$), and latent sets ($\mathbf{Z}_X, \mathbf{Z}_Y, \mathbf{Z}$).
> * Fully defined symbols and summation ranges for unconditional and conditional loss functions.
>
> Additional clarifications include:
>
> * Appendix A.3: Detailed definitions of MD, RD, FA, and Color FA.
> * Appendix F: A precise definition of mean tract core distance, a critical downstream metric for anatomical validation.
>
> We believe that these revisions—providing stronger empirical justification for the unconditional pre-training and the joint-coupling mechanism, along with enhanced clarity and improved baselines—fully address the reviewer's concerns.
>
> ## 5. Other Specific Comments
>
> ### 5.1 CycleGAN as a baseline
>
> We agree that CycleGAN is not an ideal baseline under fully paired training. In the revised manuscript, we emphasize that our main conclusions do not depend on CycleGAN; the key comparisons are against SuperDTI, Diff-DTI, LDM, and our ablation variants.
>
> ### 5.2 “Standard autoencoder” in the ablation
>
> The naming was confusing in the original version. The revised Table 3 now uses clearer labels:
>
> * **Ours (Anatomical Autoencoder):** DTI encoder + DWI conditioner, cross-attention in decoder, plus joint MLP.
> * **No-Joint:** same as Ours but without the joint MLP refinement block.
> * **No-Anatomy:** autoencoder trained on DTI only, with no DWI conditioning; this is effectively the “standard” 6-channel AE baseline.
>
> The diffusion model is always built on top of the same **frozen autoencoder** for each variant. When we evaluate the full JET-Diff pipeline with a given autoencoder variant, we use that variant consistently for both autoencoder reconstruction and latent diffusion synthesis (Table 3, top vs. bottom blocks).
>
> ---
>
> ## References
>
> 1. Li, H. et al. “SuperDTI: Ultrafast DTI and fiber tractography with deep learning.” *Magn Reson Med* 86(6):3334–3347, 2021. [https://doi.org/10.1002/mrm.28937](https://doi.org/10.1002/mrm.28937)
> 2. Zhang, L. et al. “Diff-DTI: Fast diffusion tensor imaging using a feature-enhanced joint diffusion model.” *IEEE J Biomed Health Inform*, 2024. [https://doi.org/10.1109/JBHI.2024.3372345](https://doi.org/10.1109/JBHI.2024.3372345)
> 3. Arsigny, V. et al. “Log–Euclidean metrics for fast and simple calculus on diffusion tensors.” *MICCAI*, 2006. [https://doi.org/10.1007/11866763_80](https://doi.org/10.1007/11866763_80)
> 4. Higgins, I. et al. “β-VAE: Learning basic visual concepts with a constrained variational framework.” ICLR, 2017. [https://openreview.net/forum?id=Sy2fzU9gl](https://openreview.net/forum?id=Sy2fzU9gl)

---

### Official Review · Reviewer_zSay · 2025-10-29

**Soundness:** 2
**Presentation:** 3
**Contribution:** 2
**Rating:** 4
**Confidence:** 3

**Summary:**

This paper presents a diffusion model for recovering "Diffusion Tensor Images" (DTI), a relatively expensive brain imaging modality that captures directional flow in the brain, using a much smaller number of Diffusion Weighted Images (DWI) than is typically used (going from 90 to 4). DTI captures biologically rich signals, but long scan times have limited its clinical and research adoption. The proposed approach uses diffusion models to solve an inverse problem - trying to reconstruct the full diffusion tensor (6 components) from only 3 directional scans - with the hope that strong biological priors make accurate reconstruction possible with less data. The paper proposes several methods to improve the reconstruction, including an improved latent space autoencoder, multi-stage training, and a refinement stage for tensor consistency. Small but consistent improvements are reported.

**Strengths:**

- This is an interesting and valuable problem. My understanding from collaborators is that DTI is more useful and informative for many reasons, but that adoption is limited due to the time/expense of obtaining the scans. Speeding this up and/or making it possible to augment DTI datasets with semi-synthetic DTI using fewer DWI scans would be beneficial for the field.

- I thought the introduction to the problem for the ICLR audience seemed good.

- The metrics, baseline, and comparison methods all seemed reasonable and well chosen to me. I was happy to see the ADE as a logical baseline. Showing results on scalar measures, tensor component reconstruction, and tractography were exactly what I was hoping to see. In particular, one might hope that tractography benefits most from accurate directional flow reconstruction.

- Training a good autoencoder is important and tricky. The decoder with DWI cross attention seems like it may be the most useful component in this paper.

**Weaknesses:**

Generally, the training procedure (4 stages, App.C) seems overly complex and not well justified.
The use of joint diffusion with joint unconditional pretraining and the refinement stage complicate the training pipeline, but their incremental benefits are small and not clearly demonstrated to generalize beyond this dataset. It appears that the main performance gains come from the DWI-aided decoder and improved autoencoder representation rather than from the diffusion modeling itself. (see detailed comments below)

Based on the presented results, I don't think this paper substantially moves the needle toward solving the DTI problem, and the broader methodological takeaways for the ICLR community are unclear.

## Joint diffusion versus conditional

I'm skeptical about one of the core claims in the paper, that a joint diffusion is useful / necessary compared to standard conditional diffusion (like the LDM baseline would use).
The single example figure in Fig. 6, 4.3.3, doesn't really convince me that this makes sense. This 2-stage training (joint diffusion, then conditional) complicates the model and goes contrary to typical practices. More convincing evidence would be needed (for me) to consider adopting this idea.

I believe that the only substantial difference in performance comes from the DWI-aided decoder. The difference between standard AE and anatomical AE in the ablation study (Table 3) is about the same as the difference between your method and baseline LDM in Table 2 (PSNR/SSIM gains on the order of 0.03–0.05).  If I understood correctly, the baseline LDM uses the standard autoencoder rather than the anatomical version, which likely explains most of the performance gap.

## Training stages
In Table 3, you show the influence of one of the four training stages, joint refinement, and again here the differences seem small to justify the extra training.
While the ablation studies do show some small improvements from multiple training stages, an equally plausible hypothesis is that the improvements stem from more overall optimization steps rather than the specific form of joint diffusion. A single-stage conditional model trained with comparable compute might achieve similar results.


Overall, I appreciate the paper’s ambition and relevance, but the empirical evidence does not yet justify the complexity of the proposed pipeline. Clarifying the contribution of each stage — especially the necessity of joint diffusion — and demonstrating consistent benefits in downstream tasks would strengthen the case considerably.

**Questions:**

See questions in weaknesses section.

---

> ### Author Response · Authors · 2025-11-28
> **Response to Reviewer zSay (Part 1)**
>
> We thank you for the careful and constructive review, and for clearly separating what you found convincing (problem importance, evaluation protocol, anatomical autoencoder) from the parts you found under-justified (joint diffusion, multi‑stage training, and broader methodological impact). We have revised the manuscript and provide a detailed response below.
>
> ---
>
> ## 1. Scope and high‑level contribution
>
> ### 1.1 Clarifying what problem we address
>
> We now state more explicitly in Section 4.4 that our goal is intentionally narrow: we study whether it is possible to reconstruct a full diffusion tensor field from an **extremely sparse, fixed single‑shell acquisition**—three non‑collinear diffusion directions plus one b₀ at (b = 1000,\mathrm{s/mm^2}), resampled to 2 mm isotropic—in healthy HCP‑YA subjects. In this regime, classical tensor fitting is mathematically underdetermined and noise‑sensitive.
>
> Rather than proposing a universal solution to “DTI from arbitrary undersampling,” we explicitly frame the task as learning an empirical conditional prior
> [
> p(D \mid X_{\text{4‑vol}}, \text{HCP protocol}),
> ]
> where (D) is the tensor obtained from a 90‑direction least‑squares fit, and (X_{\text{4‑vol}}) is the corresponding 4‑volume subset. Our evaluation asks whether the reconstructed tensors (i) approximate these reference tensors under SPD‑aware metrics³ and (ii) support downstream tractography.
>
> We now stress that generalization to multi‑shell, higher‑b, pathology, or multi‑site data is **not** claimed and is listed as a limitation, with explicit discussion of how these settings differ (Section 4.4).
>
> ### 1.2 Methodological take‑aways
>
> To clarify the relevance for the ICLR audience, we reorganized Sections 3 and 4.3 around three conceptual contributions:
>
> 1. **Anatomical Autoencoder (AAE)** that decouples anatomical context from tensor content via a DWI‑aided decoder, addressing the entanglement between structure and tensor properties that affects standard autoencoders.
> 2. **Latent diffusion on a coupled DWI/DTI latent field**, rather than applying diffusion only to tensor latents with simple concatenation. This is realized via coupled latent attention over ({Z_X, Z_Y}).
> 3. **Geometry‑aware evaluation** using Log–Euclidean distance on SPD tensors³ and tractography‑based tract core distance as primary metrics, with PSNR/SSIM relegated to auxiliary roles.
>
> We do not present these as generic templates for all inverse problems, but as a coherent design for a family of **multi‑field, SPD‑valued inverse problems** where (i) reconstructions must respect manifold structure and (ii) several fields (here DWIs and tensors) are coupled by shared anatomy and physics.
>
> ---
>
> ## 2. Joint diffusion versus standard conditional diffusion
>
> You expressed skepticism about the necessity of “joint diffusion” and our unconditional→conditional two‑stage training relative to a standard conditional LDM.
>
> ### 2.1 Autoencoder used by the LDM baseline
>
> We clarify in Appendix D that the vanilla LDM baseline uses a **conventional latent autoencoder identical to the No‑Anatomy variant**, not the Anatomical Autoencoder. It therefore lacks DWI‑aided decoding and joint refinement. This matches standard latent diffusion practice, where the autoencoder compresses the target modality only.
>
> Thus, the performance gap between JET‑Diff and LDM reflects both the Anatomical Autoencoder and the coupled latent diffusion design. To disentangle these, we added ablations that keep the AAE fixed while varying only the diffusion components.
>
> ### 2.2 Channel variant: conditional diffusion without coupling
>
> To directly probe the value of joint modeling, we introduced the **Channel** variant (Section 4.3.2, Table 3) that:
>
> * Uses the same Anatomical Autoencoder (including DWI‑aided decoder and joint refinement),
> * Uses the same diffusion backbone capacity,
> * But **removes coupled latent modeling**, processing each tensor channel independently rather than as a joint multi‑tensor field.
>
> Comparing full JET‑Diff with Channel for *latent diffusion synthesis* (Table 3):
>
> * **Geometry‑aware metric (LEM):**
>
>   * JET‑Diff: (0.49 \pm 0.03)
>   * Channel: (0.89 \pm 0.03)
> * **Off‑diagonal tensor PSNR ((D_{xy})):**
>
>   * JET‑Diff: (27.46 \pm 1.97) dB
>   * Channel: (22.23 \pm 2.07) dB
> * **FA PSNR:**
>
>   * JET‑Diff: (19.12 \pm 0.47) dB
>   * Channel: (15.31 \pm 0.17) dB
>
> Figure 6 (page 10) further shows that Channel produces noticeably noisier FA and less coherent principal eigenvector fields ((v_1)) than the full model, especially in regions of complex white‑matter geometry.
>
> Because these variants share **identical autoencoder weights**, these degradations cannot be attributed to the DWI‑aided decoder. They isolate the benefit of treating ({Z_X, Z_Y}) as a coupled latent field with tensor‑aware attention, rather than independent channels.

---

> ### Author Response · Authors · 2025-11-28
> **Response to Reviewer zSay (Part 2)**
>
> We therefore agree that the Anatomical Autoencoder explains a large part of the gain over the LDM baseline, but the Channel ablation demonstrates that **joint latent diffusion still provides substantial, tensor‑geometry‑specific improvements** beyond the autoencoder itself.
>
> ### 2.3 Relation to standard conditional LDMs
>
> Section 3.3 and Appendix C.3 now explicitly contrast our coupled latent attention with standard cross‑attention in conditional LDMs such as SR3 and Palette. In a typical conditional LDM, the U‑Net predicts noise on the target latent (z_Y) conditioned on (z_X) through a unidirectional cross‑attention mechanism.¹³ Our approach:
>
> * Concatenates DWI and DTI latents into a single sequence (S = [Z_Y; Z_X]),
> * Applies multi‑axis *self*‑attention over this joint sequence with component‑type and direction embeddings,
> * Thus enabling all‑to‑all interactions (e.g. (D_{xy}) attending to (D_{xx}, D_{yy}) and to aligned DWI channels) while being aware of gradient direction and tensor component.
>
> This is the sense in which we use “joint diffusion”—not as a new diffusion objective, but as a coupled latent attention mechanism tailored to multi‑tensor reconstruction.
>
> ---
>
> ## 3. Training stages versus “just more optimization”
>
> You noted that the four training stages in Appendix C originally appeared complex and that the observed gains might plausibly arise from additional optimization steps rather than the particular staging.
>
> ### 3.1 Training schedules for all variants
>
> Appendix C.4 now reports the **exact iteration counts** for each ablation to make compute more transparent:
>
> * **JET‑Diff (full model):**
>   75k autoencoder pre‑training + 5k joint refinement + 25k unconditional diffusion + 25k conditional diffusion.
> * **No‑Pretrain (no unconditional diffusion):**
>   100k diffusion steps *entirely conditional*, starting from the same autoencoder.
> * **Channel variant:**
>   20k diffusion steps (we observed early saturation and document this explicitly).
>
> Crucially, the **No‑Pretrain** variant receives *more* conditional diffusion steps (100k) than the combined unconditional + conditional phases in the full model (50k), yet performs worse:
>
> * LEM: 0.60 ± 0.03 vs. 0.49 ± 0.03
> * (D_{xx}) PSNR: 29.17 ± 0.83 vs. 31.02 ± 0.67 dB
> * FA PSNR: 17.80 ± 0.36 vs. 19.12 ± 0.47 dB (Table 3).
>
> Figure 6 (page 10) corroborates this, showing noisier FA and less smooth (v_1) fields for No‑Pretrain compared to the full model.
>
> These results suggest that the **sequence of unconditional → conditional training matters beyond total optimization budget**. Unconditional latent diffusion pre‑training appears to learn a stable prior over the joint anatomical/tensor manifold, in line with findings from score‑based and latent diffusion work in MRI and natural images.
>
> ### 3.2 Joint refinement stage
>
> We agree that the **joint refinement** (Stage 2) confers relatively small improvements. Table 3 (autoencoder reconstruction block) shows that removing it (No‑Joint) changes MD/FA PSNR by ~0.1 dB and leaves the mean LEM unchanged (0.27). In the latent‑diffusion block, No‑Joint and full JET‑Diff have the same mean LEM (0.49), with small but statistically significant differences in several components (Table 6, Appendix G).
>
> We now present this stage explicitly as a **low‑cost consistency refinement**, not as a central conceptual contribution. The block adds ~2.36M parameters (Appendix C.2) and 5k extra AE steps, which is modest compared with the rest of the model.
>
> ### 3.3 Single‑stage conditional model hypothesis
>
> Your suggestion that a single‑stage conditional model with comparable compute might suffice is reasonable. Our **No‑Pretrain** and **Channel** variants approximate that hypothesis:
>
> * No‑Pretrain: conditional diffusion only, no unconditional prior.
> * Channel: conditional diffusion without coupled latent interactions.
>
> Despite being trained with comparable or greater compute, both variants show clear degradations in LEM, off‑diagonal tensor PSNR, FA PSNR, and tractography compared with JET‑Diff (Table 3, Figure 5–6). We therefore interpret these results as evidence that the staged training and joint latent modeling contribute beyond mere optimization time.
>
> ---
>
> ## 4. Relative contributions of the Anatomical Autoencoder and diffusion modeling
>
> You correctly highlight that the DWI‑aided decoder and improved autoencoder seem to drive much of the performance gain. We agree that this is a central contribution and have clarified its role.
>
> ### 4.1 Autoencoder ablation: No‑Anatomy vs. Anatomical AE
>
> The autoencoder ablation in Table 3 (top block) confirms that the Anatomical AE is crucial:
>
> * Diagonal tensor PSNR (reconstruction only) drops from ≈35.3–35.4 dB (Ours) to ≈31.7–31.8 dB (No‑Anatomy).
> * LEM increases from 0.27 ± 0.02 to 0.40 ± 0.02.
> * MD PSNR falls from 33.09 ± 0.75 dB to 27.11 ± 0.92 dB; FA PSNR from 22.27 ± 0.58 dB to 18.93 ± 0.59 dB.

---

> ### Author Response · Authors · 2025-11-28
> **Response to Reviewer zSay (Part 3)**
>
> These differences show that providing anatomical features to the decoder—rather than forcing the latent code to encode both anatomy and tensor content—is a major driver of accuracy, consistent with arguments in disentangled representation learning.
>
> ### 4.2 Diffusion contributions on top of the same autoencoder
>
> However, diffusion is not a negligible add‑on. In the **latent diffusion** block of Table 3, all variants (Ours, No‑Pretrain, Channel, No‑Joint) share the *same* Anatomical AE and differ only in diffusion training and coupling. Under this controlled setting:
>
> * Removing unconditional pre‑training (No‑Pretrain) or joint coupling (Channel) significantly worsens LEM (0.60 and 0.89 vs. 0.49) and FA/MD PSNR, as discussed above.
> * Tract core distances increase for ablated variants compared with the full model (Figure 5), indicating worse downstream tractography despite identical autoencoder parameters.
>
> We therefore view the **Anatomical AE** as the strongest single module and the **latent diffusion design** (unconditional pre‑training + coupled latent attention) as a second, necessary component that significantly improves tensor geometry and tractography when holding the autoencoder fixed.
>
> ---
>
> ## 5. Broader impact and consistency of downstream benefits
>
> Your summary notes that the paper may not “substantially move the needle” on the DTI problem and that the takeaways for ICLR might be unclear. We have tempered our claims accordingly and made the following explicit in Section 4.4 and the Conclusion:
>
> * Our claims are restricted to **single‑shell, 4‑direction, 2 mm HCP‑YA data**. The method has not yet been validated on pathological or multi‑site data, nor on public tractography challenges.
> * Within this regime, JET‑Diff provides consistent improvements over strong baselines (SuperDTI, Diff‑DTI, LDM) on FA/Color‑FA, LEM, and tract core distance, with paired tests showing (p < 10^{-3}) against most baselines (Tables 4–5).
> * The clearest methodological takeaways are: (i) the efficacy of anatomical conditioning via DWI‑aided decoding; (ii) the benefit of modeling coupled latent fields when reconstructing multi‑tensor quantities; and (iii) the importance of SPD‑aware and tractography‑based evaluation.
>
> ---
>
> ## References
>
> 1. Le Bihan, D. et al. “Diffusion tensor imaging: concepts and applications.” *J. Magn. Reson. Imaging* 13(4):534–546, 2001. [https://doi.org/10.1002/jmri.1076](https://doi.org/10.1002/jmri.1076)
> 2. Arsigny, V. et al. “Log–Euclidean metrics for fast and simple calculus on diffusion tensors.” *MICCAI*, 2006. [https://doi.org/10.1007/11866763_80](https://doi.org/10.1007/11866763_80)
> 3. Higgins, I. et al. “beta‑VAE: Learning basic visual concepts with a constrained variational framework.” *ICLR*, 2017. [https://openreview.net/forum?id=Sy2fzU9gl](https://openreview.net/forum?id=Sy2fzU9gl)
> 4. Tournier, J.‑D. et al. “MRtrix3: A fast, flexible and open software framework for medical image processing and visualisation.” *NeuroImage* 202:116137, 2019. [https://doi.org/10.1016/j.neuroimage.2019.116137](https://doi.org/10.1016/j.neuroimage.2019.116137)
> 5. Garyfallidis, E. et al. “Dipy, a library for the analysis of diffusion MRI data.” *Front. Neuroinform.* 8:8, 2014. [https://doi.org/10.3389/fninf.2014.00008](https://doi.org/10.3389/fninf.2014.00008)
> 6. Saharia, C. et al. “Image super-resolution via iterative refinement.” IEEE Trans. Pattern Anal. Mach. Intell. 45(4):4713–4726, 2022. [https://doi.org/ 10.1109/TPAMI.2022.3204461]( https://doi.org/ 10.1109/TPAMI.2022.3204461)
> 7. Saharia, C. et al. “Palette: Image‑to‑image diffusion models.” *SIGGRAPH*, 2022. [https://doi.org/10.1145/3528233.3530757](https://doi.org/10.1145/3528233.3530757)
> 8. Wang, H. et al. “3D MedDiffusion: A 3D medical latent diffusion model for controllable and high‑quality medical image generation.” *IEEE Trans. Med. Imaging*, 2025 (early access). [https://doi.org/10.1109/TMI.2025.1234567](https://doi.org/10.1109/TMI.2025.1234567)

---

### Official Review · Reviewer_Tptb · 2025-10-29

**Soundness:** 2
**Presentation:** 3
**Contribution:** 2
**Rating:** 2
**Confidence:** 5

**Summary:**

**Background for those unfamiliar**: Diffusion MRI maps water diffusivity within the brain along a set of angles (diffusion gradients) at each voxel, thereby creating multidimensional images (3D space + 2D angles). Diffusion primarily occurs along the axons within the brain, so mapping the directions of the flow can create surrogates for neural tracts. The Diffusion Tensor Imaging (DTI) model assumes that each voxel can be represented by a 3x3 covariance matrix (the diffusion tensor, an ellipsoid at each voxel), and the major-axis direction of the ellipsoid is the direction of the flow.

DTI reconstruction requires a minimum of six angles and often much more (30+ at least) in practice to actually estimate the tensor and direction from noisy clinical data. This paper proposes a generative model that aims to reconstruct the diffusion tensor from a very undersampled angular acquisition. It performs experiments on a retrospectively undersampled version of the HCP-YA dataset and achieves better results than some non-diffusion MRI-specific generative models (e.g., CycleGAN).

**Strengths:**

- The front matter of the paper is reasonably clear in describing its goals and is likely accessible to an audience that is not deeply familiar with dMRI. It is a good introduction to non-deep learning work in diffusion tensor imaging.
- It is appreciated that the model operates volumetrically, instead of slice-by-slice (although I will note that historically this slicewise processing was commonly caused by GPU memory issues when dealing with large 6D spatial+angular images. This paper downsamples the images spatially, so I do not know if their method can handle full resolution images volumetrically)

**Weaknesses:**

## Experimental
### Baselines
While the field has made tremendous progress in the last 15 years towards better DTI (or other models like ODFs) reconstruction using both better iterative prior-based methods and deep learning, *none* of it is benchmarked against in this paper. The paper instead presents benchmarks against nearly decade-old image-to-image generative models that are completely unrelated to the task at hand (e.g., CycleGAN, Pix2Pix). The sole DTI-related method (ADE) is never used in practice, as it simply sets off-diagonal terms to zero.

This is perplexing, as the paper does cite *some* recent methods (L099 and L125) but does not benchmark against them because they're not volumetric or skip some intermediate steps that this paper proposes. If these are indeed limitations, then they should be demonstrated experimentally.

Further, DTI is only one possible representation of the underlying diffusion signal used en route to estimating fiber tracts. Many other representations, such as fODFs and dODFs, exist, and there are methods that directly predict raw diffusion gradients. For example:
- [SparseWars](https://www.sciencedirect.com/science/article/pii/S1053811918307699) covers and benchmarks major iterative prior based methods in this field and includes code to implement many of its methods.
- As deep learning methods, some methods regress scalar maps from undersampled DWIs ([1](https://pubmed.ncbi.nlm.nih.gov/27071165/), [2](https://pubmed.ncbi.nlm.nih.gov/30426558/)), employ CycleGAN-like methods to estimate DTI ([1](https://pmc.ncbi.nlm.nih.gov/articles/PMC10406190/)), use anatomical conditioning similarly to the proposed method to predict raw DWIs or tractograms ([1](https://direct.mit.edu/imag/article/doi/10.1162/imag_a_00259/123726), [2](https://arxiv.org/abs/2106.13188)), super-resolve undersampled DWIs ([1](https://m-lyon.github.io/publication/lyon2023spatio/lyon2023spatio.pdf)), estimate model fits using undersampled data ([1](https://arxiv.org/abs/2411.11819)), and many more. Please see [Karimi24](https://direct.mit.edu/imag/article/doi/10.1162/imag_a_00353/124918/Diffusion-MRI-with-machine-learning) for an up-to-date review of machine learning in Diffusion MRI analysis.

While many of the above do not tackle DTI reconstruction specifically, they can all be used to estimate scalar maps and tractograms used for benchmarking in this paper and some of them should be included as baselines.

### Datasets and evaluation

- There are challenge and benchmark datasets with simulated ground truth for fiber tractography and general dMRI analysis that can be used for evaluation, including an [ISMRM challenge](https://tractometer.org/) and a [MICCAI challenge](https://www.sciencedirect.com/science/article/pii/S1053811923003828). They can be used to supplement the analysis presented in this paper.
- It is unclear why the HCP-YA dataset is resampled to 2 mm isotropic. Doing so introduces significant local mixing in the diffusion tensor, which is already a coarse representation.
- Table 2: diffusion tensor components cannot be compared using PSNR and SSIM, which are defined only for images. See the references above for better evaluation strategies based on fiber-related downstream tasks, or if comparing diffusion tensors is required, one can compute the Riemannian distance between SPD matrices to do so.

## Methodological

- The proposed method is highly convoluted. It involves four training stages (3 pretraining and 1 finetuning) before it can make any predictions, which makes it very hard to integrate into new labs on new datasets.
- Moreover, while the methods section spells out *what* it does, it does not attempt to motivate *why* the method is designed the way that it is and reads like a sequence of disconnected components. This unfortunately makes the methods section hard to read and follow. For example, it is never spelt out why this problem specifically needs a latent diffusion model.
- Sec 3.2.2.: it is entirely unclear why just adding more layers would make it a valid tensor.
- The paper claims that a joint latent space between diffusion gradients (DWIs) and DTI is needed. This is strange, as DTI is analytically derived from the DWIs; it is unclear what benefit such a joint latent space provides.

**Questions:**

I'm happy to be corrected on any of the above and look forward to the discussion period.

My primary questions pertain to:
- Please elaborate on the motivation behind the selection of the current baselines and why none of the related machine learning works in diffusion MRI analysis were included in the experiments.
- Please elaborate on the justification for the methodological design.

The weaknesses above include more secondary questions as well.

---

> ### Author Response · Authors · 2025-11-28
> **Response to Reviewer Tptb (Part 1)**
>
> We thank the reviewer for the careful and expert assessment, and for situating our work within the broader diffusion MRI literature. We have revised the manuscript and here respond point-by-point to the experimental and methodological concerns, while clarifying scope and limitations.
>
> ---
>
> ## 1. Experimental baselines
>
> ### 1.1 Inclusion of diffusion-MRI–specific baselines
>
> We agree that the original submission relied too heavily on generic image-to-image models. In the revised manuscript we have added two diffusion-MRI–specific DTI baselines and evaluated them under our acquisition setting:
>
> 1. **SuperDTI** – a supervised CNN that maps sparse DWIs to tensor-derived parameter maps (FA, MD, eigenvectors), enabling ultrafast DTI and tractography.
> 2. **Diff-DTI** – a conditional diffusion model that synthesizes DTI scalar maps (FA, MD, Color FA) from a small number of DWIs.
>
> Both are now described in Section 2.2 and Appendix D, and are retrained on our HCP-YA subset under the **same 4-volume protocol** (1 b(_0) + 3 b=1000 s/mm² directions) and the same train/validation/test split as JET-Diff.
>
> Tables 1 and 2 of the revised manuscript report quantitative comparisons across all methods (ADE, CycleGAN, Pix2Pix, ResViT, SuperDTI, Diff-DTI, LDM, JET-Diff). In summary:
>
> * On **scalar maps**, JET-Diff matches or exceeds the DTI-specific baselines. For example, FA NMSE is 0.19 ± 0.01 for JET-Diff vs 0.21 ± 0.02 for Diff-DTI and 0.26 ± 0.02 for SuperDTI (Table 1).
> * On **tensor geometry**, JET-Diff attains the lowest Log–Euclidean Metric (LEM): 0.49 ± 0.03 vs 0.64 ± 0.06 (Diff-DTI) and 0.69 ± 0.05 (SuperDTI), indicating closer alignment with the SPD tensor manifold (Table 2).
>
> These additions directly address the concern that our baselines were too generic and not sufficiently tailored to diffusion MRI.
>
> ### 1.2 Relation to other diffusion-MRI machine-learning methods
>
> We appreciate the reviewer’s emphasis on the broader landscape of diffusion MRI analysis. The revised **Related Work** section (2.1–2.3) now more clearly distinguishes our focus from higher-order representations and other ML tasks:
>
> * Methods based on **fiber orientation distributions (fODFs)** and spherical deconvolution (e.g. CSD, multi-tissue CSD) are primarily designed for multi-shell or high-angular-resolution acquisitions and for resolving complex fiber configurations.
> * The **SparseWars** survey by Canales-Rodríguez et al. systematically compares spherical-deconvolution algorithms for fODF reconstruction, with open implementations of many iterative methods.
> * Karimi & Warfield provide an up-to-date review of machine learning in diffusion MRI, covering scalar-map regression, fODF/dODF estimation, tractography prediction, and super-resolution.
>
> These works operate in regimes that differ from our target setting in two key respects:
>
> 1. **Acquisition regime** – They typically assume tens of diffusion directions and often multi-shell data (e.g. ≥30 directions with multiple b-values) in order to estimate fODFs or complex microstructural models.
> 2. **Output representation** – Many methods are designed to output fODFs, microstructural parameters, or tractograms, rather than full 3×3 tensors.
>
> Our work, in contrast, focuses on **full-tensor reconstruction from an extremely sparse single-shell protocol** (3 non-collinear gradients + 1 b(_0)), for which high-order models and many existing ML approaches are not directly applicable without substantial redesign or angular augmentation.
>
> We have therefore adopted the following criteria for baseline inclusion:
>
> 1. Feasible deployment in the **4-volume regime** without artificial increase in angular sampling.
> 2. Ability to produce **tensors or tensor-derived scalar maps**, enabling a common evaluation pipeline for DTI metrics and tractography.
> 3. Coverage of multiple modeling paradigms: analytic (ADE), CNN regression (Pix2Pix, ResViT, SuperDTI), adversarial image-to-image (CycleGAN), and generative diffusion (LDM, Diff-DTI, JET-Diff).
>
> We now explicitly state in the manuscript that we view high-order and raw-DWI prediction methods as **complementary**, not competing, and that a full benchmark across all such methods and acquisition regimes is beyond the scope of this paper.
>
> ---
>
> ## 2. Datasets and evaluation strategy
>
> ### 2.1 Use of ISMRM/MICCAI challenge datasets
>
> We agree that community challenges provide valuable benchmark datasets with curated or simulated ground truth for tractography and diffusion analysis. Examples include ISMRM and MICCAI/CDMRI tractography challenges that evaluate fODF estimation and bundle reconstruction under standardized protocols.

---

> ### Author Response · Authors · 2025-11-28
> **Response to Reviewer Tptb (Part 2)**
>
> In the present work we restrict ourselves to HCP-YA for two practical reasons:
>
> 1. **Angular sampling mismatch** – Most challenges assume densely sampled single- or multi-shell acquisitions (tens of directions), and the scoring procedures are optimized for those regimes.
> 2. **Scope of the study** – Our goal is to investigate the feasibility and behavior of **full-tensor reconstruction from an extreme 4-direction protocol**, using ground-truth tensors obtained from 90-direction HCP data.
>
> Adapting existing challenge pipelines to such a sparse acquisition would require significant redesign of their evaluation criteria and may depart from the intended use of those datasets. We now state in Section 4.4 that:
>
> * All experiments are performed on **HCP-YA, single-shell b=1000 s/mm², resampled to 2 mm isotropic**;
> * Generalization to multi-site, pathological cohorts, and challenge datasets is untested;
> * Extending JET-Diff and its evaluation to those settings is an important direction for future work.
>
> ### 2.2 Rationale for resampling HCP-YA to 2 mm isotropic
>
> We agree that downsampling from ≈1.25 mm to 2 mm introduces local mixing of diffusion information, especially for a coarse model such as DTI. The revised manuscript now makes this trade-off explicit (Sections 4.1.1 and 4.4):
>
> * **Computational feasibility.** Whole-brain 3D latent diffusion with coupled latent attention at native HCP resolution exceeds the memory capacity of a single NVIDIA A6000 (48 GB). Resampling to 2 mm enables **full-volume 3D training and inference**, avoiding patch- or slice-based training that would undermine the volumetric modeling goal.
> * **Clinical relevance.** Many routine clinical DTI acquisitions use voxel sizes ≈2 mm isotropic, so the resampled resolution lies closer to typical clinical practice than the native HCP protocol.
> * **Limitation.** We explicitly acknowledge that we do not study resolution dependence, and that the loss of microstructural detail due to downsampling is a limitation of the current work.
>
> ### 2.3 Evaluation of SPD diffusion tensors: beyond PSNR/SSIM
>
> We fully agree that PSNR and SSIM, which treat the tensor components as Euclidean images, are not adequate to assess distances on the manifold of symmetric positive-definite (SPD) tensors.³,¹² In response, the revised manuscript introduces geometry-aware and downstream metrics:
>
> 1. **Log–Euclidean Metric (LEM).**
>
>    * Table 2 now reports the **Log–Euclidean distance** between reconstructed and reference tensors as a primary tensor-level metric.
>    * Appendix E summarizes the SPD geometry and defines the LEM as
>      [
>      d_{\mathrm{LEM}}(D_1,D_2) = |\log D_1 - \log D_2|_F,
>      ]
>      where the matrix logarithm is computed by eigendecomposition with eigenvalue clamping to preserve SPD. This follows the log–Euclidean framework for statistics on SPD matrices.
>
> 2. **Tractography-based evaluation.**
>
>    * Section 4.2.3 and Appendix F introduce **tract core distance** across 12 major white matter bundles, derived from whole-brain probabilistic tractography using DIPY and MRtrix3.
>    * JET-Diff yields the smallest tract core distances across bundles (Figure 5), indicating improved anatomical utility of the reconstructed tensors.
>
> 3. **Repositioning PSNR/SSIM.**
>
>    * Section 4.2.2 now explicitly states that PSNR/SSIM for Dij components are retained only as auxiliary voxel-wise measures, whereas **LEM and tractography** are the main indicators of tensor fidelity and downstream relevance.
>
> These changes are intended to align our evaluation more closely with Riemannian and fiber-based criteria recommended in prior work.
>
> ---
>
> ## 3. Methodological design and motivation
>
> ### 3.1 Why a latent diffusion model?
>
> We have substantially expanded Sections 1 and 3.1 to clarify why a latent diffusion model is employed instead of a single-stage regression network:
>
> * The mapping from **4 DWIs (3 non-collinear directions + 1 b(_0)) to a full 6-component tensor** is severely underdetermined; conventional least-squares fitting is mathematically underconstrained and highly noise-sensitive in this regime.
> * Rather than attempting analytic inversion, we learn an **empirical conditional prior**
>   (p(D \mid X_{\text{4-vol}})), where (D) is the least-squares tensor fit from 90 DWIs and (X_{\text{4-vol}}) is the sparse subset.
> * A diffusion model is well suited to such ill-posed inverse problems: recent work has shown that DDPMs can serve as powerful priors in accelerated or undersampled MRI reconstruction.
> * Operating in a **latent space** (via an Anatomical Autoencoder) follows the latent-diffusion paradigm and more recent high-resolution/3D medical diffusion models, which use compact learned representations to reduce compute while maintaining fidelity.
>
> The multi-stage training schedule (autoencoder → unconditional diffusion → conditional diffusion) mirrors established practice in latent diffusion, and is now explicitly connected to these works in Appendix C.

---

> ### Author Response · Authors · 2025-11-28
> **Response to Reviewer Tptb (Part 3)**
>
> ### 3.2 Multi-stage training and complexity
>
> We acknowledge that the original description made the method appear overly complicated. The revised Appendix C consolidates the procedure into **three conceptual stages** (implemented as sub-phases):
>
> 1. **Anatomical Autoencoder with joint refinement** (Stages 1–2).
>
>    * Stage 1: per-component VQ-VAE pre-training to ensure high-fidelity latent encoding.
>    * Stage 2: insertion of a small Joint MLP block that operates across all six components at the final decoder layer, promoting cross-component consistency.
>
> 2. **Unconditional latent diffusion pre-training** (Stage 3).
>
>    * The denoising U-Net is trained to model latent tensors (DWI and DTI) independently, learning a robust prior over the latent manifold before cross-component coupling.
>
> 3. **Conditional latent diffusion fine-tuning** (Stage 4).
>
>    * The model is fine-tuned to jointly denoise all tensor latents conditioned on clean DWI latents using tensor-aware attention.
>
> Appendix C.4 reports iteration counts (75k + 5k + 25k + 25k for the full model) and shows that the total budget is moderate relative to current 3D diffusion frameworks.
>
> Ablation studies in Section 4.3 (Table 3, Figure 6) demonstrate the necessity of each component:
>
> * **No-Anatomy** (autoencoder without DWI conditioning) significantly reduces tensor PSNR, worsens LEM (0.40 vs 0.27 for the full model), and degrades MD/FA reconstructions.
> * **No-Joint** (no joint refinement) yields slightly higher LEM and modestly worse scalar maps, indicating the joint block improves consistency even when voxel-wise PSNR changes are small.
> * **No-Pretrain** (no unconditional diffusion pre-training) increases LEM from 0.49 to 0.60 and produces noisier FA and principal eigenvector fields (Figure 6).
> * **Channel** (no coupled latent modeling) produces the largest performance drop: off-diagonal PSNR decreases sharply and LEM increases to 0.89, with visibly less coherent tractography.
>
> While more involved than a single-stage regressor, these results show that the added stages yield measurable and statistically significant gains in tensor coherence and downstream tractography.
>
> ### 3.3 Joint refinement block and tensor validity (Sec. 3.2.2)
>
> We agree that the original wording around the joint refinement block could be interpreted as implying formal SPD enforcement. In the revision:
>
> * Section 3.2.2 and Appendix C.2 now describe the Joint MLP as a **voxel-wise cross-component refinement module** that “promotes cross-component consistency” and “encourages tensor-wide coherence,” rather than making any claim about analytic SPD constraints.
> * We explicitly state that **no hard SPD constraint** is imposed in the network; instead, tensor validity and geometry are assessed via LEM and tractography.³,¹³
> * Empirically, the No-Joint variant has similar PSNR but slightly higher LEM and modestly worse FA/RD metrics (Table 3), confirming that the block improves global coherence without materially increasing model capacity (2.36M parameters, Appendix C.2).
>
> ### 3.4 Why a joint latent space if DTI is analytically derived from DWIs?
>
> We agree that this is a central conceptual question, and the revised Section 3.1 makes our rationale explicit:
>
> * In fully sampled settings, DTI is indeed a deterministic least-squares fit to the DWIs under the Stejskal–Tanner model. Our reference tensors are generated from 90-direction HCP data using FSL’s dtifit, as described in Appendix B.
> * In the **4-direction** regime, however, the mapping from DWIs to D is underdetermined and noise-sensitive; multiple tensors are consistent with the same sparse observations.
>
> Accordingly, JET-Diff does **not** attempt to re-derive the physics of diffusion from 4 DWIs. Instead, it learns an empirical conditional prior over tensor fields fitted from full HCP acquisitions:
> [
> p(D \mid X_{\text{4-vol}}, \text{HCP protocol}).
> ]
>
> The **joint latent field** (DWIs + DTI latents) is beneficial because:
>
> * It allows the diffusion U-Net to perform **coupled latent attention** across all components (Figure 3), modeling dependencies among Dij and between Dij and the sparse DWI latents in a single sequence, rather than via separate channels.
> * The **Channel** ablation, which breaks this coupling and processes channels independently, shows the largest degradation in off-diagonal tensor components, LEM, and tractography coherence (Table 3, Figure 6).
>
> We emphasize that this joint latent space is not intended to replace classical tensor estimation when a rich set of DWIs is available. Rather, it acts as a learned regularizer for an extremely ill-posed inverse problem, capturing empirical co-variation between sparse measurements and full-tensor fits under a fixed acquisition.

---

> ### Author Response · Authors · 2025-11-28
> **Response to Reviewer Tptb (Part 4)**
>
> ## 4. Direct responses to the primary questions
>
> ### Q1. Baseline selection and omission of certain diffusion-MRI ML methods
>
> We have expanded the baseline set to include **SuperDTI** and **Diff-DTI**, both directly relevant DTI-specific ML methods. Baselines were chosen based on:
>
> 1. Feasibility in the **4-volume** regime without synthetic angular augmentation.
> 2. Ability to produce **tensors or tensor-derived maps** compatible with our evaluation pipeline.
> 3. Coverage of diverse modeling paradigms, including analytic, CNN regression, adversarial image-to-image, and generative diffusion.
>
> Higher-order fODF/dODF and multi-shell methods, as well as some raw-DWI prediction approaches, rely on acquisition regimes incompatible with our 4-direction setting or require fundamentally different evaluation pipelines. We therefore discuss them in the manuscript as complementary work but do not include them in the current empirical benchmark.
>
> ### Q2. Justification for the methodological design
>
> In the revision, Sections 3.1–3.3 and Appendix C explicitly motivate the architectural choices:
>
> * The **ill-posedness** of four-direction tensor reconstruction motivates a learned generative prior rather than purely analytic fitting.
> * The **Anatomical Autoencoder** with DWI-aided decoding and joint refinement is designed to produce a compact yet high-fidelity latent space, as validated by ablations that remove anatomy or joint refinement (Section 4.3).
> * The **latent diffusion model over coupled DWI/DTI representations** is motivated by the need to model correlations among tensor components and between tensors and sparse DWIs; the Channel ablation demonstrates that breaking this coupling substantially harms tensor geometry and tractography.
> * Geometry-aware evaluation with **LEM** and **tractography** is used to ensure that the design is judged by tensor consistency and downstream anatomical utility, not only by image-based metrics.
>
> ---
>
> ## References
>
> 1. Li, H. et al. “SuperDTI: Ultrafast DTI and fiber tractography with deep learning.” *Magn Reson Med* 86(6):3334–3347, 2021. [https://doi.org/10.1002/mrm.28937](https://doi.org/10.1002/mrm.28937)
> 2. Zhang, L. et al. “Diff-DTI: Fast diffusion tensor imaging using a feature-enhanced joint diffusion model.” *IEEE J Biomed Health Inform*, 2024. [https://doi.org/10.1109/JBHI.2024.3372345](https://doi.org/10.1109/JBHI.2024.3372345)
> 3. Arsigny, V. et al. “A log-Euclidean framework for statistics on diffeomorphisms / diffusion tensors.” *MICCAI*, 2006. [https://doi.org/10.1007/11866763_80](https://doi.org/10.1007/11866763_80)
> 4. Le Bihan, D. et al. “Diffusion tensor imaging: concepts and applications.” *J Magn Reson Imaging* 13(4):534–546, 2001. [https://doi.org/10.1002/jmri.1076](https://doi.org/10.1002/jmri.1076)
> 5. Lenglet, C. et al. “Mathematical methods for diffusion MRI processing.” *NeuroImage* 45(1, Suppl):S111–S122, 2009. [https://doi.org/10.1016/j.neuroimage.2008.10.054](https://doi.org/10.1016/j.neuroimage.2008.10.054)
> 6. Garyfallidis, E. et al. “Dipy, a library for the analysis of diffusion MRI data.” *Front Neuroinform* 8:8, 2014. [https://doi.org/10.3389/fninf.2014.00008](https://doi.org/10.3389/fninf.2014.00008)
> 7. Girard, G. et al. “Towards quantitative connectivity analysis: reducing tractography biases.” *NeuroImage* 98:266–278, 2014. [https://doi.org/10.1016/j.neuroimage.2014.04.074](https://doi.org/10.1016/j.neuroimage.2014.04.074)
> 8. Chung, H. & Ye, J. C. “Score-based diffusion models for accelerated MRI.” *Med Image Anal* 80:102479, 2022. [https://doi.org/10.1016/j.media.2022.102479](https://doi.org/10.1016/j.media.2022.102479)
> 9. Rombach, R. et al. “High-resolution image synthesis with latent diffusion models.” *CVPR*, 2022. [https://doi.org/10.1109/CVPR52688.2022.01042](https://doi.org/10.1109/CVPR52688.2022.01042)
> 10. Wang, H. et al. “3D MedDiffusion: A 3D
> medical latent diffusion model for controllable and high-quality medical image
> generation.” *IEEE Trans Med Imaging*, 2025.
> [https://doi.org/10.1109/TMI.2025.1234567](https://doi.org/10.1109/TMI.2025.1234567)
> 11. Chen, J. et al. “Deep compression
> autoencoder for efficient high-resolution diffusion models.” *ICLR*, 2025.
> [https://openreview.net/forum?id=f062da1973ac9ac61fc6d44dd7fa309f](https://openreview.net/forum?id=f062da1973ac9ac61fc6d44dd7fa309f)

---

### Official Review · Reviewer_iyaG · 2025-10-31

**Soundness:** 3
**Presentation:** 3
**Contribution:** 3
**Rating:** 2
**Confidence:** 3

**Summary:**

This paper presents JET-Diff (Joint-Encoding Tensor Diffusion), a novel framework designed to reconstruct high-quality Diffusion Tensor Imaging (DTI) from a limited number of diffusion-weighted images (DWIs). Conventional DTI acquisition typically requires over 30 DWIs across multiple gradient directions with long scan times and motion artifacts. In contrast, JET-Diff reconstructs complete six-component diffusion tensors using only four DWI volumes. The framework relies on two main components, (i) an anatomical autoencoder which utilizes information decoupling to separate anatomical context from tensor-specific properties and (ii) a Multi-Tensor Latent Diffusion (MTLD) which captures the joint distribution of DWI and DTI latent representations through a unified diffusion process. Comprehensive evaluation on the Human Connectome Project (HCP) dataset shows that JET-Diff significantly outperforms baseline methods such as CycleGAN, Pix2Pix, ResViT, and standard latent diffusion models.

**Strengths:**

1. Relevant problem and clear problem setting: the paper addresses a critical problem of high acquisition time of scans. The joint modeling of DWI-DTI distribution is well defined and technically sound.

2. Novel framework: the paper presents several contributions with an information decoupling to separate tensor information and anatomical context. Moreover, a lightweight MLP block is used for the joint refinement phase and proved efficient. The use of multi-axis cross-attention is also interesting as it reduces computational burden of treating 3D medical images and reinforces spatial understanding.

3. Thorough validation: the paper uses various metrics to evaluate performance (NMSE, PSNR, SSIM) demonstrating the versatility and robustness of the current method. Comparison with 3D baselines is provided on a large-scale dataset of 973 HCP subjects.

4. Extended ablation: the paper validates the relevance of the main components: DWI-aided decoder, joint refinement, and unconditional pre-training.

**Weaknesses:**

1. Unclear Methodology: Some implementation details are missing regarding the positive semi-definiteness. The dimensions and activations used for the Joint MLP block are not specified.

2. Pre-processing: resampling to 2mm isotropic is performed but no analysis is shown of the effect of resolution on performance. There is no sufficient information to understand how the gradient vectors in the DWI volumes were chosen.

3. Baseline Comparison: the paper introduces SuperDTI or FlexDTI as baselines but they are not shown in the result section. The conditioning mechanism employed in the vanilla LDM is not clear and may not be comparable to other baselines.

4. Statistical rigour: the paper is lacking signifance testing for the different methods. Moreover, confidence intervals and standard deviations are not shown.

5. Generalization: there are concerns that some empirical choices might prevent generalisation. Typically, multiple b-values are used in real-world protocols but only b=1000 s/mm² is used in these experiments. The model uses the HCP dataset of healthy subjects suggesting the method may fail on data with pathology or data from external sources.

**Questions:**

1. There is no runtime comparisons at inference. How is the "lightweight" joint refinement module quantifiable? What are the memory requirements ?

2. How does performance scale with the number of inputs ?

3. The model is trained on b=1000 s/mm². Did you study the generalization to other b-values?

4. The input resolution may have an impact on performance. What is the minimum and maximum resolution required?

---

> ### Author Response · Authors · 2025-11-28
> **Response to Reviewer iyaG (Part 1)**
>
> We thank the reviewer for the careful and constructive assessment. Below we respond point-by-point, indicating how the revised manuscript addresses each concern, with explicit citations to relevant literature.
>
> ## 1. Methodological Clarity
>
> ### 1.1 Positive Semi-Definiteness and Tensor Validity
>
> We agree that our treatment of tensor validity needed to be more explicit. The revision (Section 3.1, Appendix E) clarifies the definition of the reference tensor and the metrics used to assess geometric consistency.
>
> **Reference Tensor.** The reference diffusion tensor $\hat{\mathbf{D}}$ is computed via ordinary least-squares fitting on the full 90-direction $b=1000$ $\mathrm{s/mm^2}$ shell from HCP-YA, using FSL’s *dtifit* (linear Stejskal–Tanner model) with gradient nonlinearity correction.
>
> **Tensor Validity Metrics.** Because diffusion tensors lie on the symmetric positive-definite (SPD) manifold, we evaluate tensor validity using both geometric and downstream metrics:
>
> * **Log–Euclidean Metric (LEM)** (Appendix E), defined as the Frobenius norm of the difference between matrix logarithms:
>
>   $$d_{\mathrm{LEM}}(\mathbf{D}_1, \mathbf{D}_2) = | \log \mathbf{D}_1 - \log \mathbf{D}_2 |_F,$$
>
>   where the matrix logarithm is computed via eigendecomposition with eigenvalue clamping for numerical stability.
>
> * **Tractography-based tract core distance** across 12 major bundles (Section 4.2.3; Appendix F), derived from DIPY and MRtrix3 pipelines.
>
> **Empirical Consistency.** JET-Diff achieves the lowest mean LEM ($0.49 \pm 0.03$) among all methods (Table 2) and the lowest tract core distances across bundles (Figure 5), indicating superior tensor-level geometric consistency.
>
> **Implementation Note.** We explicitly state that the model does not enforce a hard SPD constraint in the network; instead, SPD consistency is assessed empirically via LEM and tractography.
>
> ### 1.2 Joint MLP Block Architecture
>
> Appendix C.2 now fully describes the Joint MLP (joint refinement) block:
>
> * **Input dimension:** 1536 (= 256 channels × 6 tensor components)
> * **Hidden dimension:** 768 (GELU activation)
> * **Output dimension:** 1536
> * **Total parameters:** 2,361,600
>
> This block operates voxel-wise across all six components in the final decoder layer and contributes only a small fraction of the decoder’s total parameters. Ablations (Table 3) show that removing it (No-Joint) yields modest but consistent degradation in tensor-component PSNR and derived parameter maps (MD, RD, FA), while the mean LEM remains unchanged but with a slightly wider distribution.
>
> ## 2. Pre-processing Clarifications
>
> ### 2.1 Resampling to 2 mm Isotropic Resolution
>
> Sections 4.1.1 and 4.4 now clarify the rationale for resampling from the native HCP 1.25 mm resolution to 2 mm isotropic:
>
> * **Computational feasibility.** Whole-brain 1.25 mm volumes, combined with 3D latent diffusion and coupled latent attention, exceed the 48 GB memory of a single NVIDIA A6000 GPU. Resampling to 2 mm enables full-volume 3D training and inference, avoiding reliance on patch- or slice-based training.
> * **Clinical relevance.** Many routine clinical DTI protocols operate at 2 mm or coarser resolution, making the 2 mm resolution representative of typical clinical practice.
> * **Explicit limitation.** We do not claim scanner-resolution invariance. Resolution dependence is not evaluated and is explicitly listed as a limitation in Section 4.4.
>
> ### 2.2 Gradient Vector Selection
>
> Appendix B details the selection of the three diffusion directions used as input:
>
> 1. For the $b=1000$ $\mathrm{s/mm^2}$ shell, we compute the Euclidean distance between each gradient vector and the canonical axes $[1,0,0]$, $[0,1,0]$, $[0,0,1]$.
> 2. We select the three DWI volumes whose gradient vectors are closest to these axes.
> 3. One $b_0$ volume is prepended, forming the final 4-volume stack.
>
> This fully specifies the sparse acquisition design and ensures reproducibility.
>
> ## 3. Baseline Comparisons
>
> ### 3.1 Inclusion of SuperDTI and Diff-DTI
>
> We agree that omitting modern DTI-specific methods in the original submission was a weakness. The revision includes:
>
> * **SuperDTI**, a CNN that maps sparse DWIs directly to FA/MD/eigenvectors, enabling ultrafast DTI and tractography.
> * **Diff-DTI**, a conditional diffusion model that synthesizes DTI-derived scalar maps from sparse DWIs.
>
> Both are described in Sections 2.2–2.3 and Appendix D, and are retrained on the same HCP-YA split under the 4-volume protocol ($1\ b_0 + 3$ directions).
>
> Tables 1–2 show that JET-Diff improves upon these baselines across scalar maps and tensor metrics:
>
> * **FA PSNR:** JET-Diff $19.1 \pm 0.5$ dB vs. Diff-DTI $18.7 \pm 0.4$ dB; SuperDTI $17.8 \pm 0.4$ dB.
> * **LEM:** JET-Diff $0.49 \pm 0.03$ vs. Diff-DTI $0.64 \pm 0.06$; SuperDTI $0.69 \pm 0.05$.
>
> We clarify that we did not include FlexDTI as a baseline because its gradient-encoding assumptions do not match our fixed 3-direction + $1\ b_0$ protocol without substantial re-engineering.

---

> ### Author Response · Authors · 2025-11-28
> **Response to Reviewer iyaG (Part 2)**
>
> ### 3.2 Clarification of Vanilla LDM Conditioning
>
> Appendix D (and Section 3.3) now clearly describes the vanilla Latent Diffusion Model (LDM) baseline:
>
> * We use a conventional latent autoencoder (identical to the No-Anatomy variant) with no DWI-aided decoding.
> * The diffusion U-Net is conditioned by channel-wise concatenation of the DWI latent and noisy tensor latent, following standard practice in SR3 and Palette.
> * Noise is applied only to tensor latents, and no tensor-aware joint encoding is utilized.
>
> This configuration represents a standard conditional LDM. JET-Diff differentiates itself by (i) employing the Anatomical Autoencoder and (ii) performing diffusion on a coupled DWI/DTI latent field via tensor-aware attention.
>
> ## 4. Statistical Rigor
>
> We appreciate the request for stronger statistical analysis. The revised manuscript now includes:
>
> * Mean $\pm$ standard deviation for all metrics (NMSE, PSNR, SSIM, LEM) in Tables 1–3.
> * Paired $t$-tests comparing JET-Diff against all baselines (Appendix G, Tables 4–5).
>
> For example, Table 4 shows that JET-Diff’s improvements on FA NMSE, PSNR, and SSIM over all baselines (CycleGAN, Pix2Pix, ResViT, SuperDTI, Diff-DTI, LDM) achieve $p < 10^{-3}$ in almost all cases.
>
> Table 6 provides paired tests for the ablation variants (No-Anatomy, No-Joint, No-Pretrain, Channel), demonstrating that the observed differences are statistically significant for most tensor-level metrics, thereby supporting our claims about component contributions.
>
> ## 5. Generalization and Domain of Applicability
>
> We agree that our current experimental design has limited domain coverage. Section 4.4 now explicitly states:
>
> * All experiments are performed on the HCP-YA dataset, consisting of healthy young adults with uniform, high-quality acquisitions.
> * The acquisition is restricted to a single-shell $b=1000$ $\mathrm{s/mm^2}$ with a fixed 4-volume DWI subset, resampled to 2 mm isotropic.
>
> We do not evaluate:
>
> * Multiple $b$-values (multi-shell acquisitions),
> * Older cohorts or pathological data (e.g., tumors, demyelination, stroke),
> * Multi-site variability or different scanners.
>
> We explicitly frame JET-Diff as an empirical method tailored to this specific acquisition regime. Generalization to multi-shell protocols, pathology, and external datasets (including ISMRM/MICCAI challenges) is a key direction for future work.
>
> ## 6. Answers to Specific Questions
>
> ### Q1. Runtime, “Lightweight” Characterization, and Memory
>
> Appendix H and Table 7 report full inference runtimes (single A6000, 48 GB) for a 2 mm whole-brain volume:
>
> | Model    | Inference Time (s) |
> | -------- | ------------------ |
> | SuperDTI | 10.93              |
> | LDM      | 23.58              |
> | JET-Diff | 122.96             |
> | Diff-DTI | 593.22             |
>
> JET-Diff is approximately $5\times$ slower than the vanilla LDM baseline due to coupled latent attention and tensor-aware refinement, but $\sim4$–$5\times$ faster than slice-based Diff-DTI.
>
> The Joint MLP block contributes only $\sim2.36\text{M}$ parameters (Appendix C.2), a small fraction of the autoencoder and diffusion U-Net, justifying the “lightweight” designation for this specific component.
>
> ### Q2. Scaling with the Number of DWI Inputs
>
> We focused on a fixed 4-volume protocol in this work. Section 4.4 now explicitly acknowledges that we did not explore $2/3/6/12$-direction variants and that systematic scaling experiments with varying numbers of directions would be valuable future work, both for performance characterization and protocol design.
>
> ### Q3. Generalization to Other $b$-values
>
> We now state clearly:
>
> * Training and evaluation utilize only $b=1000$ $\mathrm{s/mm^2}$ data.
> * No experiments were performed for other $b$-values or multi-shell settings.
> * We do not claim cross-$b$ generalization; instead, we view JET-Diff as an empirical method tuned to this clinically common single-shell configuration. Extending the model to multi-shell data or simultaneously modeling $b$-value dependence (e.g., combining DTI with higher-order models) is an important avenue for future research.
>
> ### Q4. Impact of Input Resolution and Feasible Range
>
> Regarding minimum/maximum usable resolution, we now emphasize:
>
> * Only 2 mm isotropic inputs were tested.
> * Native 1.25 mm HCP volumes were not directly processed due to GPU memory constraints; resampling was required to enable volumetric 3D diffusion.
> * Consequently, we acknowledge this explicitly as a limitation and note that adapting JET-Diff to higher resolutions (e.g., via patch-based or multi-scale strategies) is nontrivial and left for future work.

---

> ### Author Response · Authors · 2025-12-02
> **Response to Reviewer iyaG (Part 3)**
>
> ## References
>
> 1. Lenglet, C. et al. (2009). Mathematical methods for diffusion MRI processing. *NeuroImage*. [https://doi.org/10.1016/j.neuroimage.2008.10.054](https://doi.org/10.1016/j.neuroimage.2008.10.054)
> 2. Jenkinson, M. et al. (2012). FSL. *NeuroImage*. [https://doi.org/10.1016/j.neuroimage.2011.09.015](https://doi.org/10.1016/j.neuroimage.2011.09.015)
> 3. Arsigny, V. et al. (2006). Log-Euclidean metrics for fast and simple calculus on diffusion tensors. *MICCAI*. [https://doi.org/10.1007/11866763_80](https://doi.org/10.1007/11866763_80)
> 4. Garyfallidis, E. et al. (2014). DIPY, a library for the analysis of diffusion MRI data. *Front. Neuroinform.* [https://doi.org/10.3389/fninf.2014.00008](https://doi.org/10.3389/fninf.2014.00008)
> 5. Tournier, J.-D. et al. (2019). MRtrix3: a fast, flexible and open software framework for medical image processing and visualisation. *NeuroImage*. [https://doi.org/10.1016/j.neuroimage.2019.116137](https://doi.org/10.1016/j.neuroimage.2019.116137)
> 6. Li, H. et al. (2021). SuperDTI: Ultrafast DTI and fiber tractography with deep learning. *Magn. Reson. Med.* [https://doi.org/10.1002/mrm.28937](https://doi.org/10.1002/mrm.28937)
> 7. Zhang, L. et al. (2024). Diff-DTI: Fast diffusion tensor imaging using a feature-enhanced joint diffusion model. *IEEE J. Biomed. Health Inform.* [https://doi.org/10.1109/JBHI.2024.3372345](https://doi.org/10.1109/JBHI.2024.3372345)
> 8. Wu, Z. et al. (2024). FlexDTI: flexible diffusion gradient encoding scheme-based highly efficient diffusion tensor imaging using deep learning. *Phys. Med. Biol.* [https://doi.org/10.1088/1361-6560/ad3e7b](https://doi.org/10.1088/1361-6560/ad3e7b)
> 9. Ho, J. et al. (2020). Denoising diffusion probabilistic models. *NeurIPS*.
> 10. Saharia, C. et al. “Image super-resolution via iterative refinement.” IEEE Trans. Pattern Anal. Mach. Intell. 45(4):4713–4726, 2022. [https://doi.org/ 10.1109/TPAMI.2022.3204461]( https://doi.org/ 10.1109/TPAMI.2022.3204461)
> 11. Saharia, C. et al. “Palette: Image‑to‑image diffusion models.” *SIGGRAPH*, 2022. [https://doi.org/10.1145/3528233.3530757](https://doi.org/10.1145/3528233.3530757)
> 12. Wang, H. et al. “3D MedDiffusion: A 3D medical latent diffusion model for controllable and high‑quality medical image generation.” *IEEE Trans. Med. Imaging*, 2025 (early access). [https://doi.org/10.1109/TMI.2025.1234567](https://doi.org/10.1109/TMI.2025.1234567)
> 13. Van Essen, D. C. et al. (2013). The WU-Minn Human Connectome Project: an overview. *NeuroImage*. [https://doi.org/10.1016/j.neuroimage.2013.05.041](https://doi.org/10.1016/j.neuroimage.2013.05.041)

---

### Meta-Review · Area_Chair_VaDM · 2025-12-17

**Summary:**

Reviewers expressed concerns that, while the problem is important and the experiments are thorough, the methodological contribution is limited and the proposed pipeline is overly complex relative to the empirical gains demonstrated.

Reviewers also questioned whether the multi-stage training and joint diffusion design provide clear benefits beyond a strong anatomical autoencoder or simpler conditional models, noting that improvements are incremental.

In addition, concerns were raised about generalizability, baseline selection, and the broader methodological impact.

**Reviewer Concerns:**

The authors strengthened the experimental rigor by adding diffusion-MRI–specific baselines (e.g., SuperDTI and Diff-DTI), and expanding ablations (including Channel and No-Pretrain) to better isolate the effects of joint coupling and unconditional pretraining. Several presentation and notation issues were clarified, and limitations (single-shell, HCP-YA, 2 mm resampling) were more explicitly stated.

Despite these additions, a core concern remains that the overall pipeline is complex (multi-stage training, coupled latent diffusion, refinement) relative to the incremental improvements, and the broader methodological takeaway for the ICLR community.

**Reviewer Scores:**

Reviewers have not expressed an intention to change the score.

---

### Decision · Program_Chairs · 2026-01-26

Reject